# Optimal Regret Is Achievable with Bounded Approximate Inference Error: An Enhanced Bayesian Upper Confidence Bound Framework

[1]**Ziyi Huang,** [1]**Henry Lam,** [2]**Amirhossein Meisami,** [1]**Haofeng Zhang** *
[1] Columbia University, New York, NY, USA
[2] Adobe Inc., San Jose, CA, USA
zh2354,khl2114,hz2553@columbia.edu, meisami@adobe.com

## Abstract

Bayesian bandit algorithms with approximate Bayesian inference have been widely used in real-world applications. However, there is a large discrepancy between the superior practical performance of these approaches and their theoretical justification. Previous research only indicates a negative theoretical result: Thompson sampling could have a worst-case linear regret $\Omega(T)$ with a constant threshold on the inference error measured by one $\alpha$-divergence. To bridge this gap, we propose an Enhanced Bayesian Upper Confidence Bound (EBUCB) framework that can efficiently accommodate bandit problems in the presence of approximate inference. Our theoretical analysis demonstrates that for Bernoulli multi-armed bandits, EBUCB can achieve the optimal regret order $O(\log T)$ if the inference error measured by two different $\alpha$-divergences is less than a constant, regardless of how large this constant is. To our best knowledge, our study provides the first theoretical regret bound that is better than $o(T)$ in the setting of constant approximate inference error. Furthermore, in concordance with the negative results in previous studies, we show that only one bounded $\alpha$-divergence is insufficient to guarantee a sub-linear regret.

## 1 Introduction

The stochastic bandit problem, dated back to [37], is an important sequential decision-making problem that aims to find optimal adaptive strategies to maximize cumulative reward. At each time step, the learning agent chooses an action among all possible actions and observes its corresponding reward (but not others), and thus requires a balance between exploration and exploitation. Previous theoretical studies mainly focus on *exact* Bayesian bandit problems, requiring access to exact posterior distributions. However, their work cannot be easily applied to complex models such as deep neural networks, where maintaining exact posterior distributions tends to be intractable [36]. In contrast, *approximate* Bayesian bandit methods are widely employed in real-world applications with state-of-the-art performances [36, 41, 33, 45, 17, 50]. In comparison with exact algorithms, approximate Bayesian bandit algorithms are more challenging to analyze, as the inaccessibility of exact posterior sampling adds another level of discrepancy, and the resulting theory and solutions hence also differ substantially.

Few theoretical studies have been developed around Bayesian bandit approaches with approximate inference, despite their superior practical performance. [28] gave a theoretical analysis of an approximate sampling method called Ensemble Sampling, which possessed constant Kullback–Leibler divergence (KL divergence) error from the exact posterior and thus indicated a linear regret. [34]

---

*Authors are listed alphabetically.

37th Conference on Neural Information Processing Systems (NeurIPS 2023).

showed that with a *constant* threshold on the inference error in terms of $\alpha$-divergence (a generalization of KL divergence), regardless of how small the threshold is, Thompson sampling with general approximate inference could have a linear regret in the worst case. Their work also showed that Thompson sampling combined with a small amount of forced exploration could achieve a $o(T)$ regret upper bound, but no better result than $o(T)$ was shown. Moreover, this improvement was mostly credited to the forced exploration rather than the intrinsic property of Thompson sampling. It appears that [34] illustrated a paradox that approximate Bayesian bandit methods worked well empirically but failed theoretically. Thus, further study regarding the fundamental understanding of approximate Bayesian bandit methods is necessary.

Motivated by the negative results in [34], [30] leveraged an efficient Markov chain Monte Carlo (MCMC) Langevin algorithm in multi-armed Thompson sampling so that the inference error would *vanish* along with increased sample numbers. [48] extended [30] to the contextual bandit problems and integrated contextual Thompson sampling with the Langevin algorithm that allowed the approximate posterior distribution to be sufficiently close to the exact posterior distribution. Hence, both works [48, 30] had a similar feature in terms of vanishing inference error. However, in other inference approaches, such as variational inference [8], the approximate posteriors might incur a systematic computational bias. To accommodate the latter scenario, we consider a general Bayesian inference approach that allows bounded inference error:

*Is it possible to achieve the optimal regret order $O(\log T)$ with a **constant** (non-vanishing) threshold on the inference error?*

This question is not well investigated in previous literature, even for Bernoulli bandits. [34] showed that the answer is *No* for Thompson sampling when the inference error measured by only *one* $\alpha$-divergence is bounded. In this study, we will provide a novel theoretical framework and point out that the answer could be *Yes* when the inference error measured by *two* different $\alpha$-divergences is bounded where one $\alpha$ is greater than 1, and the other $\alpha$ is less than 0. This assumption guarantees that the approximate posterior is close to the exact posterior from two different "directions". Our finding implies that the problem of sub-optimal regret in the presence of approximate inference may not arise from the constant but from the design of the inference error. Our study takes the first step in deriving positive answers in the presence of constant approximate inference error, which provides some theoretical support for the superior performance of approximate Bayesian bandit methods.

In this study, we extend the work of the Bayesian Upper Confidence Bound (BUCB) [21, 20, 42, 17] to the setting of approximate inference and propose an enhanced Bayesian bandit algorithm that can efficiently accommodate approximate inference, termed as Enhanced Bayesian Upper Confidence Bound (EBUCB). In particular, we redesign the quantile choice in the algorithm to address the challenge of approximate inference: The original choice of $t^{-1}$ provides the best regret bound without approximate inference, but in the presence of approximate inference, it leads to an undesirable quantile shift which degrades the performance. By adjusting the quantile choice, we theoretically demonstrate that EBUCB can achieve the optimal regret order $O(\log T)$ if the inference error measured by two different $\alpha$-divergences ($\alpha_1 > 1$ and $\alpha_2 < 0$) is bounded. We also provide insights in the other direction: Instead of two different $\alpha$-divergences, controlling one $\alpha$-divergence alone is not sufficient to guarantee a sub-linear regret for both Thompson sampling, BUCB, and EBUCB. This further suggests that naive approximate inference methods that only minimize one $\alpha$-divergence alone could perform poorly, and thus it is critical to design approaches with two different $\alpha$-divergences reduced.

Our main contributions are summarized as follows:
1) We propose a general Bayesian bandit framework, named EBUCB, to address the challenge of approximate inference. Our theoretical study shows that for Bernoulli bandits, EBUCB can achieve a $O(\log T)$ regret upper bound when the inference error measured by two different $\alpha$-divergences is bounded. To the best of our knowledge, with constant approximate inference error, there is no existing result showing a regret upper bound that is better than $o(T)$, even for Bernoulli bandits.
2) We develop a novel sensitivity analysis of quantile shift with respect to inference error. This provides a fundamental tool to analyze Bayesian quantiles in the presence of approximate inference, which holds promise for broader applications, e.g., when the inference error is time-dependent.
3) We demonstrate that one bounded $\alpha$-divergence alone is insufficient to guarantee a sub-linear regret. Worst-case examples are constructed and illustrated where Thompson sampling/BUCB/EBUCB has $\Omega(T)$ regret if only one $\alpha$-divergence is bounded. Hence, special consideration on reducing two different $\alpha$-divergences is necessary for real-world applications.

4) Our experimental evaluations corroborate our theory well, showing that our EBUCB is consistently superior to BUCB and Thompson sampling on multiple approximate inference settings.

**Related Work.** Bandit problems and their theoretical optimality have been extensively studied over decades [27, 10]. The seminal paper [25] (and subsequently [11]) established the first problem-dependent frequentist regret lower bound, showing that without any prior knowledge on the distributions, a regret of order $(\log T)$ is unavoidable. Two popular lines of Bayesian bandit algorithms, Thompson sampling [1, 22, 16] and BUCB [21, 20], had been shown to match the lower bound, which indicated the theoretical optimality of those algorithms. Beyond Gaussian processes [42] and linear models [3, 39], exact computation of the posterior distribution is generally intractable, and thus, the approximate Bayesian inference techniques are necessary.

Some recent work focused on designing specialized methods to construct Bayesian indices since previous studies had demonstrated that Thompson sampling with constant inference error could exhibit linear regret in the worst-case scenario [28, 34]. [30] constructed Langevin algorithms to generate approximate samples with decreasing inference error and showed an optimal problem-dependent frequentist regret. [32] proposed variational Bayesian optimistic sampling, suggesting solving a convex optimization problem over the simplex at every time step. Unlike these researches, our study presents general results that only depend on the error threshold of approximate inference, rather than some specific approximate inference approaches.

Beyond Bayesian, another mainstream of bandit algorithms to address the exploration-exploitation tradeoff is upper confidence bound (UCB)-type frequentist algorithms [5–7, 13, 15, 40, 51, 24, 29, 43]. [12] revealed that Thompson sampling empirically outperformed UCB algorithms in practice, partly because UCB was typically conservative, as its configuration was data-independent which led to over-exploration [18]. BUCB [21] could be viewed as a middle ground between Thompson sampling and UCB. On the other hand, empirical studies [20] showed that Thompson sampling and BUCB performed similarly well in general.

## 2  Methodology

The stochastic multi-armed bandit problem consists of a set of $K$ actions (arms), each with a stochastic scalar reward following a probability distribution $\nu_i$ ($i = 1, ..., K$). At each time step $t = 1, ..., T$, where $T$ is the time horizon, the agent chooses an action $A_t \in [K]$ and in return observes an independent reward $X_t$ drawn from the associated probability distribution $\nu_{A_t}$. The goal is to devise a strategy $\mathcal{A} = (A_t)_{t \in [T]}$, to maximize the accumulated rewards through the observations from historical interactions.

In general, a wise strategy should be sequential, in the sense that the upcoming actions are determined and adjusted by the past observations: letting $\mathcal{F}_t = \sigma(A_1, X_1, ..., A_t, X_t)$ be the $\sigma$-field generated by the observations up to time $t$, $A_t$ is $\sigma(\mathcal{F}_{t-1}, U_t)$-measurable, where $U_t$ is a uniform random variable independent from $\mathcal{F}_{t-1}$ (as algorithms may be randomized). More precisely, let $\mu_j$ ($j \in [K]$) denote the mean reward of the action $j$ (i.e., the mean of the distribution $\nu_i$), and without loss of generality, we assume that $\mu_1 = \max_{j \in [K]} \mu_j$. Then maximizing the rewards is equivalent to minimizing the (frequentist) regret, which is defined as the expected difference between the reward accumulated by an "ideal" strategy (a strategy that always playing the best action), and the reward accumulated by a strategy $\mathcal{A}$:

$$R(T, \mathcal{A}) := \mathbb{E}\left[T\mu_1 - \sum_{t=1}^{T} X_t\right] = \mathbb{E}\left[\sum_{t=1}^{T}(\mu_1 - \mu_{A_t})\right]. \tag{1}$$

The expectation is taken with respect to both the randomness in the sequence of successive rewards from each action $j$, denoted by $(Y_{j,s})_{s \in \mathbb{N}}$, and the possible randomization of the algorithm, $(U_t)_{t \in [T]}$. Let $N_j(t) = \sum_{s=1}^{t} \mathbb{1}(A_s = j)$ denote the number of draws from action $j$ up to time $t$, so that $X_t = Y_{A_t, N_{A_t}(t)}$. Moreover, let $\hat{\mu}_{j,s} = \frac{1}{s} \sum_{k=1}^{s} Y_{j,k}$ be the empirical mean of the first $s$ rewards from action $j$ and let $\hat{\mu}_j(t)$ be the empirical mean of action $j$ after $t$ rounds of the bandit algorithm. Therefore $\hat{\mu}_j(t) = 0$ if $N_j(t) = 0$, $\hat{\mu}_j(t) = \hat{\mu}_{j, N_j(t)}$ otherwise.

Note that the true mean rewards $\boldsymbol{\mu} = (\mu_1, ..., \mu_K)$ are fixed and unknown to the agent. In order to perform Thompson sampling, or more generally, Bayesian approaches, we artificially define a prior distribution $\Pi_0$ on $\boldsymbol{\mu}$. Let $\Pi_t$ be the exact posterior distribution of $\boldsymbol{\mu}|\mathcal{F}_{t-1}$ with density function $\pi_t$

---
**Algorithm 1** Enhanced Bayesian Upper Confidence Bound (EBUCB) with Approximate Inference
___

**Input:** $T$ (time horizon), $\Pi_0 = Q_0$ (initial prior on $\boldsymbol{\mu}$), $c$ (parameters of the quantile), and a real-value increasing sequence $\{\gamma_t\}$ such that $\gamma_t \to 1$ as $t \to \infty$
**for** $t = 1$ **to** $T$ **do**
    **for** each action $j = 1, ..., K$ **do**
        Compute $qu_j(t) = Qu(\gamma_t, Q_{t-1,j})$.
    **end for**
    Draw action $A_t = \arg\max_{j=1,...,K} qu_j(t)$
    Get reward $X_t = Y_{A_t, N_{A_t}(t)}$
    Obtain the approximate distribution $Q_t$
**end for**
___

with marginal distributions $\pi_{t,1}, ..., \pi_{t,K}$ for actions $1, ..., K$. Specifically, if at time step $t$, the agent chooses action $A_t = j$ and consequently observes $X_t = Y_{A_t, N_{A_t}(t)}$, the Bayesian update for action $j$ is

$$\pi_{t,j}(\theta) \propto \nu_\theta(X_t)\pi_{t-1,j}(\theta), \tag{2}$$

whereas for $i \neq j$, $\pi_{t,i} = \pi_{t-1,i}$. At each time step $t$, we assume that the exact posterior computation in (2) cannot be obtained explicitly and an approximate inference method is able to give us an approximate distribution $Q_t$ (instead of $\Pi_t$). We use $q_t$ to denote the density function of $Q_t$.

First, we consider a standard case where the exact posterior is accessible. In Thompson sampling [1, 44], a sample $\hat{m}$ is drawn from the posterior distribution $\Pi_{t-1}$ and then an action $A_t$ is selected using the following strategy: $A_t = i$ if $\hat{m}_i = \max_j \hat{m}_j$. In BUCB [21], we compute the quantile of the posterior distribution $qu_j(t) = Qu(1 - \frac{1}{t(\log T)^c}, \Pi_{t-1,j})$ for each action $j$, where $Qu(\gamma, \rho)$ is the quantile function associated to the distribution $\rho$, such that $P_\rho(X \leq Qu(\gamma, \rho)) = \gamma$. Then we select action $A_t$ as follows: $A_t = i$ if $qu_i(t) = \max_j qu_j(t)$.

Next, we move to a more concrete example, Bernoulli multi-armed bandit problems with a standard setting used in seminal papers [1, 2, 21, 22]. In these problems, each (stochastic) reward follows a Bernoulli distribution $\nu_i \sim \text{Bernoulli}(\mu_i)$ and these distributions are independent of each other. The prior $\Pi_{0,j}$ is typically chosen to be the independent and identically distributed (i.i.d.) $\text{Beta}(1, 1)$, or the uniform distribution for every action $j$. Then the posterior distribution for action $j$ is a Beta distribution $\Pi_{t,j} = \text{Beta}(1 + S_j(t), 1 + N_j(t) - S_j(t))$, where $S_j(t) = \sum_{s=1}^{t} \mathbb{1}\{A_s = j\}X_t$ is the empirical cumulative reward from action $j$ up to time $t$. Then, Thompson sampling/BUCB chooses the samples/quantiles of the posterior $\Pi_{t,j} = \text{Beta}(1 + S_j(t), 1 + N_j(t) - S_j(t))$ respectively at each time step.

In the presence of approximate inference, Thompson sampling draws the sample $\hat{m}$ from $Q_{t-1}$, as the exact $\Pi_{t-1}$ is not accessible. Correspondingly, we modify the specific sequence of quantiles chosen by the BUCB algorithm with a general sequence of $\{\gamma_t\}$-quantiles and term it as Enhanced Bayesian Upper Confidence Bound (EBUCB) algorithm. The detailed pseudo algorithm of EBUCB is described in Algorithm 1. Note that the choice of $\gamma_t$ should address the presence of inference error and should be trailed to the specific definition of inference error; See Remark 3.10 in Section 3.

## 3 Theoretical Analysis

In this section, we present a theoretical analysis of EBUCB. In Section 3.1, we provide the necessary background of $\alpha$-divergence on approximate inference error measurement. Then in Section 3.2, we develop a novel sensitivity analysis of quantile shift with respect to inference error. This provides a fundamental tool to analyze Bayesian quantiles in the presence of approximate inference. The general results therein will be used for our derivation for the regret upper bound of EBUCB in Section 3.3, and are also potentially useful for broad applications, e.g., when the inference error is time-dependent. Lastly, in Section 3.4, we provide examples where Thompson sampling/BUCB/EBUCB has a linear regret with arbitrarily small inference error measured by one $\alpha$-divergence alone. All proofs are given in the Appendix.

## 3.1 The Alpha Divergence for Inference Error Measurement

The $\alpha$-divergence, generalizing the KL divergence, is a common way to measure errors in inference methods.

**Definition 3.1.** The $\alpha$-divergence between two distributions $P_1$ and $P_2$ with density functions $p_1(x)$ and $p_2(x)$ is defined as: $D_\alpha(P_1, P_2) = \frac{1}{\alpha(\alpha-1)} \left( \int p_1(x)^\alpha p_2(x)^{1-\alpha} dx - 1 \right)$, where $\alpha \in \mathbb{R}$ and the case of $\alpha = 0$ and $1$ is defined as the limit.

Note that different studies use the $\alpha$ parameter in different ways. Herein, our definition of $\alpha$-divergence does not follow Renyi's definition of $\alpha$-divergence [35]; Instead, we follow a generalized version of Tsallis's $\alpha$-divergence, which is adopted by [52, 31, 34]. Compared with Renyi's $\alpha$-divergence, Tsallis's $\alpha$-divergence does not involve a log function, and it has the following property:

**Proposition 3.2** (Positivity and symmetry)**.** *For any $\alpha \in \mathbb{R}$, $D_\alpha(P_1, P_2) \geq 0$ and $D_\alpha(P_1, P_2) = D_{1-\alpha}(P_2, P_1)$.*

The $\alpha$-divergence contains many distances such as $KL(P_2, P_1)(\alpha \to 0)$, $KL(P_1, P_2)(\alpha \to 1)$, Hellinger distance ($\alpha = 0.5$), and $\chi^2$ divergence ($\alpha = 2$). $\alpha$-divergence is widely used in variational inference [8, 23, 26], which is one of the most popular approaches in Bayesian approximate inference. Moreover, it was also adopted in previous studies on Thompson sampling with approximate inference [34, 28]. In particular, the KL divergence (($\alpha = 1$)-divergence) is: $KL(P_1, P_2) = \int p_1(x) \log \left( \frac{p_1(x)}{p_2(x)} \right) dx$. In approximate Bayesian inference, the exact posterior distribution $\Pi_t$ and the approximate distribution $Q_t$ may differ from each other. To provide a statistical analysis of approximate sampling methods, we use the $\alpha$-divergence as the measurement of inference error (statistical distance) between $\Pi_t$ and $Q_t$. Our starting point is the following:

**Assumption 3.3.** Suppose that there exists a positive value $\epsilon \in (0, +\infty)$ and two different parameters $\alpha_1 > 1$ and $\alpha_2 < 0$ such that

$$D_{\alpha_1}(Q_{t,j}, \Pi_{t,j}) \leq \epsilon, \ \forall t \in [T], j \in [K],$$
$$D_{\alpha_2}(Q_{t,j}, \Pi_{t,j}) \leq \epsilon, \ \forall t \in [T], j \in [K]. \tag{3}$$

This assumption is adapted from [34] but we enhance theirs with two bounded $\alpha$-divergences, as [34] showed one bounded $\alpha$-divergence was not sufficient to guarantee the sublinear regret. However, in the following, we show that the optimal regret order $O(\log T)$ is indeed achievable under Assumption 3.3 with two bounded $\alpha$-divergences. Intuitively, $P_2$ is flatten to cover $P_1$'s entire support when minimizing $D_\alpha(P_1, P_2)$ with a large $\alpha$ (greater than 1), while when $\alpha$ is small (less than 0), $P_2$ fits the $P_1$'s dominant mode; See [31] for the implication of $\alpha$-divergence. Therefore, Assumption 3.3 guarantees that the approximate posterior is close to the exact posterior from two different "directions". It is worth mentioning that when one $\alpha$-divergence is small, it does not necessarily imply that any other $\alpha$-divergences are large or infinite. In fact, as long as the two distributions have densities with the same support, then any $\alpha$-divergence between them is finite. Note that Assumption 3.1 does not require the threshold $\epsilon$ to be small; instead, $\epsilon$ can be any finite positive number. We pinpoint that this assumption, as well as our subsequent results, are very general in the sense that it does not depend on any specific methods of approximate inference. To enhance credibility on Assumption 3.3, we make several additional remarks in Section A.

## 3.2 Quantile Shift with Inference Error

In this section, we develop a novel sensitivity analysis of quantile shift with respect to inference error, which implies that under Assumption 3.3, the $\gamma$-quantiles of $\Pi_{t,j}$ and $Q_{t,j}$ only differs from a bound depending on $\epsilon$. We provide a general result first, which is rigorously stated as follows:

**Theorem 3.4.** *Consider any two distributions $P_1$ and $P_2$ with densities $p_1(x)$ and $p_2(x)$. Let $R_i$ denote the quantile function of the distribution $P_i$, i.e., $R_i(p) := Qu(p, P_i)$ ($i = 1, 2$). Let $0 < \gamma < 1$. Let $\delta_{\gamma,\epsilon}$ satisfy that $R_1(\gamma) = R_2(\gamma + \delta_{\gamma,\epsilon})$ where $-\gamma \leq \delta_{\gamma,\epsilon} \leq 1 - \gamma$.*

*a) If $D_\alpha(P_1, P_2) \leq \epsilon$ where $\alpha > 1$, then*

$$\delta_{\gamma,\epsilon} \leq 1 - \gamma - (\epsilon\alpha(\alpha - 1) + 1)^{\frac{1}{1-\alpha}} (1 - \gamma)^{\frac{\alpha}{\alpha-1}}.$$

*Note that when $\alpha > 1$, $(\epsilon\alpha(\alpha - 1) + 1)^{\frac{1}{1-\alpha}} < 1$ and $(1 - \gamma)^{\frac{\alpha}{\alpha-1}} < 1 - \gamma$.*

*b) If $D_\alpha(P_1, P_2) \leq \epsilon$ where $\alpha < 0$, then*

$$\delta_{\gamma,\epsilon} \geq 1 - \gamma - (\epsilon\alpha(\alpha-1) + 1)^{\frac{1}{1-\alpha}} (1-\gamma)^{\frac{\alpha}{\alpha-1}} .$$

*Note that when $\alpha < 0$, $(\epsilon\alpha(\alpha-1) + 1)^{\frac{1}{1-\alpha}} > 1$ and $(1-\gamma)^{\frac{\alpha}{\alpha-1}} > 1 - \gamma$.*

*c) Suppose that $\alpha \in (0,1)$ and $\epsilon \geq \frac{-1}{\alpha(\alpha-1)}$. Then for any $\delta_{\gamma,\epsilon} \in [-\gamma, 1 - \gamma]$, there exist two distributions $P_1$ and $P_2$ such that $D_\alpha(P_1, P_2) \leq \epsilon$. This implies that the condition $D_\alpha(P_1, P_2) \leq \epsilon$ cannot control the quantile shift between $P_1$ and $P_2$ in general when $\alpha \in (0,1)$.*

Theorem 3.4 states that $\gamma$-quantile of the distribution $P_1$ is the $(\gamma + \delta_{\gamma,\epsilon})$-quantile of the distribution $P_2$ where the quantile shift $\delta_{\gamma,\epsilon}$ has the following properties. a) The upper bound of $\delta_{\gamma,\epsilon}$ is close to 0 if $D_\alpha(P_1, P_2)$ with $\alpha > 1$ is bounded; b) The lower bound of $\delta_{\gamma,\epsilon}$ is close to 0 if $D_\alpha(P_1, P_2)$ with $\alpha < 0$ is bounded; c) A slightly large bound on $D_\alpha(P_1, P_2)$ with $0 < \alpha < 1$ cannot control the shift $\delta_{\gamma,\epsilon}$ in general, which gives the intuition that $\alpha \in (0,1)$ is not implemented in Assumption 3.3.

This theorem is distribution-free, in the sense that the bound of $\delta_{\gamma,\epsilon}$ does not depend on any specific distributions (noting that distribution changes as $t$ evolves in bandit problems). In particular, a)+b) in Theorem 3.4 shows that $C_{\epsilon,\alpha_1} (1-\gamma)^{\frac{\alpha_1}{\alpha_1-1}} \geq \delta_{\gamma,\epsilon} - (1-\gamma) \geq -C_{\epsilon,\alpha_2} (1-\gamma)^{\frac{\alpha_2}{\alpha_2-1}}$ , with $\alpha_1 > 1$ and $\alpha_2 < 0$ where $C_{\epsilon,\alpha} = (\epsilon\alpha(\alpha-1) + 1)^{\frac{1}{1-\alpha}}$ is independent of $\gamma$ or distributions, and thus independent of the time step $t$ in our EBUCB algorithm. This observation is important in the robustness of using quantiles in the EBUCB. The proof of Theorem 3.4 relies on the following lemma, which provides a quantile-based representation of $\alpha$-divergence.

**Lemma 3.5.** *Under the same conditions in Theorem 3.4, we have that for any $\alpha$-divergence,*
$D_\alpha(P_1, P_2) = \frac{\int_0^1 \left( \frac{d}{du} R_2^{-1}(R_1(u)) \right)^{1-\alpha} du - 1}{\alpha(\alpha-1)}.$

### 3.3 Finite-Time Regret Bound for EBUCB

In this section, we derive a finite-time upper bound of problem-dependent frequentist regret for our EBUCB algorithm in Bernoulli multi-armed bandit problems. Without loss of generality, we assume action 1 is optimal in the subsequent theorems.

To begin with, we first express the frequentist regret as $R(T, \mathcal{A}) := \mathbb{E}\left[ \sum_{t=1}^T (\mu_1 - \mu_{A_t}) \right] = \sum_{j=1}^K (\mu_1 - \mu_j)\mathbb{E}[N_j(t)]$ by (1). Therefore, it is sufficient to study each term $\mathbb{E}[N_j(t)]$ in order to bound the regret $R(T, \mathcal{A})$. For $(p, q) \in [0, 1]^2$, we denote the Bernoulli KL divergence between two points by $d(p, q) = p\log(\frac{p}{q}) + (1-p)\log(\frac{1-p}{1-q})$, with $0\log 0 = 0\log(0/0) = 0$ and $x\log(x/0) = +\infty$ for $x > 0$ by convention. We also denote that $d^+(p, q) = d(p, q)\mathbb{1}\{p < q\}$ for convenience.

Note that $\pi_{t,j}(x)$ is the density of Beta$(1 + S_j(t), 1 + N_j(t) - S_j(t))$ so its (closed) support is $[0, 1]$. We put a basic assumption on the approximate distribution $Q_{t,j}$.

**Assumption 3.6.** *$Q_{t,j}$ has the density $q_{t,j}(x)$ whose support is $[0, 1]$ for any $t \in [T], j \in [K]$.*

The following is our main theorem, which establishes a finite-time regret bound for our EBUCB algorithm.

**Theorem 3.7.** *Suppose Assumptions 3.3 and 3.6 hold. Let $M_{\epsilon,1} = (\epsilon\alpha_1(\alpha_1-1) + 1)^{\frac{1}{1-\alpha_1}} < 1$, $\tilde{\alpha}_1 = \frac{\alpha_1}{\alpha_1-1} > 0$, $M_{\epsilon,2} = (\epsilon\alpha_2(\alpha_2-1) + 1)^{\frac{1}{1-\alpha_2}} > 1$, and $\tilde{\alpha}_2 = \frac{\alpha_2}{\alpha_2-1} > 0$. For any $\xi > 0$, choosing the parameter $c$ such that $c\tilde{\alpha}_2 \geq 5$ in the EBUCB algorithm and setting $\gamma_t = 1 - \frac{1}{t^\zeta(\log T)^c}(\zeta > 0)$, the number of draws of any sub-optimal action $j \geq 2$ is upper-bounded by*

$$\mathbb{E}[N_j(T)] \leq \frac{\left( \zeta\tilde{\alpha}_2 + c\tilde{\alpha}_2 \right) M_{\epsilon,2} e T^{1-\zeta\tilde{\alpha}_2}}{1 - \zeta\tilde{\alpha}_2} + o(T^{1-\zeta\tilde{\alpha}_2})$$

*if $0 < \zeta\tilde{\alpha}_2 < 1$, and*

$$\mathbb{E}[N_j(T)] \leq \frac{(1+\xi)\zeta\tilde{\alpha}_1}{d(\mu_j, \mu_1)} \log(T) + o(\log T)$$

*if $\zeta\tilde{\alpha}_2 \geq 1$.*

Theorem 3.7 provides an exact *finite-time* regret bound and the $o(\cdot)$ term in Theorem 3.7 has an exact finite-time closed-form expression that holds for any time horizon $T$; See Step 4 in the proof of Theorem 3.7 in Appendix C. We only show the most dominant term of the regret bound and shrink the rest to the $o(\cdot)$ term to improve the readability of the main paper. Note that this bound has explicit dependence on $\epsilon$, which is $M_{\epsilon,1}^{-1}$ and $M_{\epsilon,2}$ in Step 4 in the proof of Theorem 3.7. Obviously, the error terms $M_{\epsilon,1}^{-1}$ and $M_{\epsilon,2}$ in the bound increase as $\epsilon$ increases. However, this dependence on $\epsilon$ does not impact the dominating term too much. The exact posterior will be more "concentrated" on the true mean with small variability as the time t increases, and the impact from the error will vanish; See Remark 3.9.

It is easy to see that to minimize the regret upper bound, we may choose $\zeta = \frac{1}{\tilde{\alpha}_2}$ in Theorem 3.7.

**Corollary 3.8.** *Under the same conditions in Theorem 3.7, for any $\xi > 0$, choosing the parameter $c$ such that $c\tilde{\alpha}_2 \geq 5$ in the EBUCB algorithm and setting $\gamma_t = 1 - \frac{1}{t^{1/\tilde{\alpha}_2}(\log T)^c}$, the number of draws of any sub-optimal action $j \geq 2$ is upper-bounded by $\mathbb{E}[N_j(T)] \leq \frac{(1+\xi)\frac{\tilde{\alpha}_1}{\tilde{\alpha}_2}}{d(\mu_j,\mu_1)}\log(T) + o(\log T)$.*

This result states that with the $\epsilon$ error threshold, the regret of the EBUCB algorithm is bounded above by $O(\log T)$ regardless of how large $\epsilon$ is, which reaches the same order $(\log T)$ of the problem-dependent frequentist regret lower bound [25]. In comparison with the exact lower bound, there is a slight difference in the multiplier before the order $(\log T)$: Our upper bound in Corollary 3.8 has the additional multiplier $\frac{\tilde{\alpha}_1}{\tilde{\alpha}_2} > 1$, which arises from the approximate inference when estimating the posterior distributions (Assumption 3.3). If in addition Assumption 3.3 holds for any $\alpha_1 > 1$ and any $\alpha_2 < 0$, then we can let $\frac{\tilde{\alpha}_1}{\tilde{\alpha}_2} \to 1$ by taking $\alpha_1 \to +\infty$ and $\alpha_2 \to -\infty$ to match the exact lower bound.

In the absence of approximate inference, [21] showed that $\mathbb{E}[N_j(T)] \leq \frac{(1+\xi)}{d(\mu_j,\mu_1)}\log(T) + o(\log T)$ matching the exact lower bound. Prior to our work, it was unclear in the literature whether the optimal regret order $O(\log T)$ could be achieved in the presence of bounded approximate inference error. Our result provides a positive answer to this question, despite the fact that the inference error may cause the multiplier $\frac{\tilde{\alpha}_1}{\tilde{\alpha}_2}$ before the order $(\log T)$. To the best of our knowledge, this is the first algorithm providing the regret upper bound that is better than $o(T)$ in [34] with bounded approximate inference error.

As discussed in [21], the horizon-dependent term $(\log T)^c$ in Corollary 3.8 is only an artifact of the theoretical analysis to obtain a finite-time regret upper bound. In practice, the model with choice $c = 0$ (i.e., without the horizon-dependent term) already achieves superior performance. This is confirmed by our experiments in Section 4. A similar observation in BUCB was indicated in [21].

*Remark* 3.9. It might appear a little surprising that the result in Corollary 3.8 indicates a regret upper bound with the dominating term $O(\log T)$ that does not depend on $\epsilon$, as one may expect that a large $\epsilon$ allows the "full swap" between the posterior of the optimal action and the posterior of a suboptimal action, making any Bayesian-based approaches unable to distinguish them. However, benefiting from historical observations, the exact posterior will be more "concentrated" on the true mean with small variability, which will keep enlarging the $\alpha$-divergence between two actions. This indicates that, for a fixed $\epsilon$, the $\alpha$-divergence between the exact posteriors of two actions can be sufficiently large as $t$ increases and ultimately exceeds $\epsilon$. Hence, the "full swap" will not happen when $t$ is large.

*Remark* 3.10. The $\frac{1}{t^{1/\tilde{\alpha}_2}}$ in EBUCB, instead of the original $\frac{1}{t}$ in BUCB, is a delicate choice to address the tradeoff between making the regret optimal without approximate inference and the presence of inference error. On a technical level, a power $\zeta$ close to 1 in $t^\zeta$ improves the regret bound without the presence of approximate inference but simultaneously leads to high-level quantile shift caused by approximate inference. Choosing $\zeta = \frac{1}{\tilde{\alpha}_2}$ is a subtle balance of these two.

The technical derivation of Theorem 3.7 depends on analyzing the quantiles of the approximate distributions used in the EBUCB algorithm. In particular, one of the major techniques in our analysis is Lemma C.1 in Appendix C. It provides explicit upper and lower bounds on the tails of approximate distributions to control the quantiles designed by the EBUCB algorithm. It is obtained by combining the quantile shift between the approximate and exact posterior distributions that developed in Theorem 3.4 (Section 3.2) with the tight bounds on the quantiles of the exact posterior distributions (the proof of Lemma 1 in [21]). This result is then used to bound the expectation of a decomposition of $N_j(T)$ in Lemma C.2 that links $N_j(T)$ to the over-estimation of the optimal arm.

### 3.4 Negative Results

We show that one bounded $\alpha$-divergence alone cannot guarantee a sub-linear regret. We provide two worst-case examples, one where Thompson sampling has a linear regret, and the other where BUCB/EBUCB has a linear regret, even when the inference error measured by one $\alpha$-divergence is small. A similar study on Thompson sampling was conducted in [34] with a special focus on the inference error on the joint distribution of all actions. In our study, nevertheless, we focus on a setting where the inference error on the distribution of each action is assumed; See Remark A.3. Therefore, the examples in [34] cannot be directly applied in our setting. Moreover, our second example shows that BUCB/EBUCB could have a linear regret if only one $\alpha$-divergence is considered, which is new.

**Assumption 3.11.** Suppose that there exists a positive value $\epsilon \in (0, +\infty)$ such that

$$D_\alpha(Q_{t,j}, \Pi_{t,j}) \leq \epsilon, \ \forall t \in [T], j \in [K]. \tag{4}$$

We establish the following theorem for Thompson sampling:

**Theorem 3.12.** *Consider a Bernoulli multi-armed bandit problem where the number of actions is $K = 2$ and $\mu_1 > \mu_2$. The prior $\Pi_{0,j}$ is chosen to be the i.i.d. $Beta(1,1)$, or the uniform distribution for every action $j = 1, 2$. For any given $\alpha < 1$ and any error threshold $\epsilon > 0$, there exists a sequence of distributions $Q_{t-1}$ such that for all $t \geq 1$:*
*1) The probability of sampling from $Q_{t-1}$ choosing action 2 is greater than a positive constant independent of $t$.*
*2) $Q_{t-1}$ satisfies Assumptions 3.6 and 3.11.*
*Therefore Thompson sampling from the approximate distribution $Q_{t-1}$ will cause a finite-time linear frequentist regret: $R(T, \mathcal{A}) = \Omega(T)$.*

This theorem shows that making one $\alpha$-divergence a small constant alone, even for each action $j$, is not sufficient to guarantee a sub-linear regret of Thompson sampling. Note that Theorem 3.12 is an enhancement of the results in [34] in the sense that the $Q_t$ constructed by our theorem satisfies more restrictive assumptions. We can derive a similar observation for the BUCB/EBUCB algorithm as follows:

**Theorem 3.13.** *Consider a Bernoulli multi-armed bandit problem where the number of actions is $K = 2$ and $\mu_1 > \mu_2$. The prior $\Pi_{0,j}$ is chosen to be the i.i.d. $Beta(1,1)$, or the uniform distribution for every action $j = 1, 2$. Consider the general EBUCB algorithm described in Algorithm 1. For any given $\alpha < 1$ and any error threshold $\epsilon > 0$, there exists a constant $T_0$ (only depending on $\epsilon$, $\alpha$, and the sequence $\{\gamma_t\}$) and a sequence of distributions $Q_{t-1}$ such that for all $t \geq 1$:*
*1) The EBUCB algorithm always chooses action 2 when $t \geq T_0$.*
*2) $Q_{t-1}$ satisfies Assumptions 3.6 and 3.11.*
*Therefore the EBUCB algorithm from the approximate distribution $Q_{t-1}$ will cause a finite-time linear frequentist regret: $R(T, \mathcal{A}) = \Omega(T)$.*

This theorem shows that making one $\alpha$-divergence a small constant alone, even for each action $j$, is insufficient to guarantee a sub-linear regret of BUCB/EBUCB. We emphasize that the examples in Theorems 3.12 and 3.13 are in the *worst-case* sense, indicating that there exist worst-case examples where Thompson sampling/EBUCB exhibits a linear regret if only one $\alpha$-divergence is bounded. However, this does not imply that EBUCB and Thompson sampling would fail on average in the presence of approximate inference. In fact, Theorem 3.7 shows that a sub-linear regret can be achieved if the inference error measured by two different $\alpha$-divergences is bounded.

## 4 Experiments

In this section, we conduct numerical experiments to show the correctness of our theory.[2] In Section 4.1, we compare the performance of EBUCB with the following baselines: BUCB (using its originally proposed quantile and $Q_t$ as $\Pi_t$ since the exact posterior distribution $\Pi_t$ is unavailable) and Thompson sampling (using $Q_t$ as $\Pi_t$). In Section 4.2, we construct worst-case examples showing that both EBUCB and Thompson sampling can degenerate to linear regret if only one $\alpha$-divergence is bounded. We consider the Bernoulli multi-armed bandit problem which has two actions with mean rewards

---

[2]The source code for experiments is available at `https://github.com/HZ0000/EBUCB`.

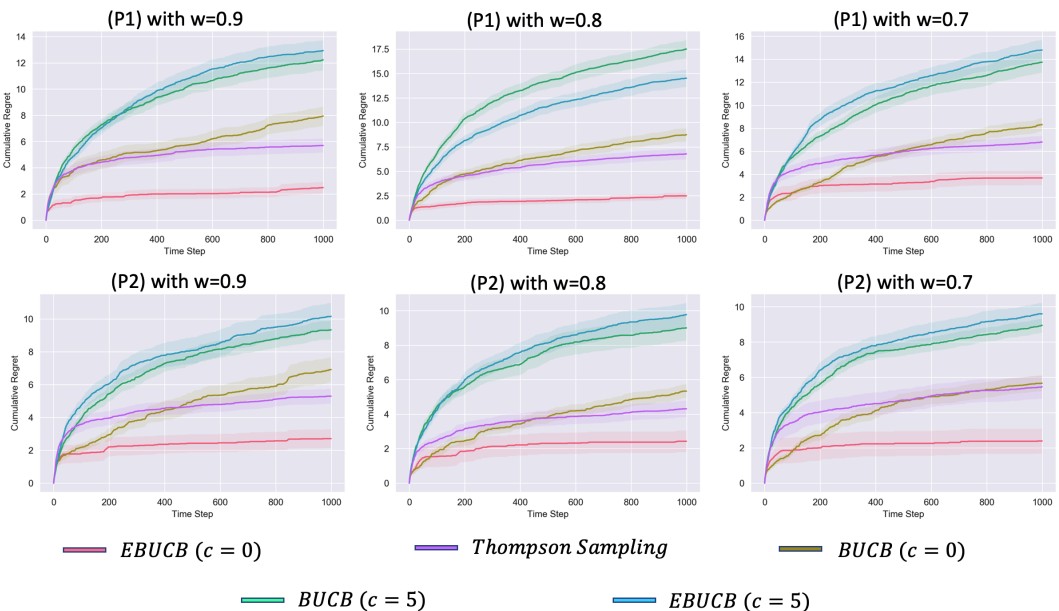

Figure 1: Comparison of EBUCB and baselines with generally misspecified posteriors under different problem settings. Results are averaged over 10 runs with shaded standard errors.

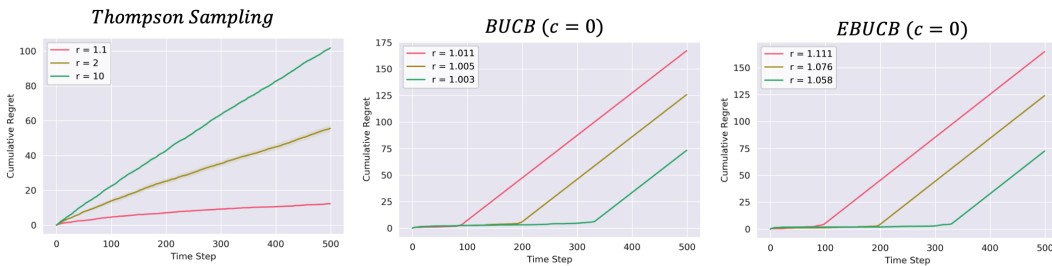

Figure 2: Results of Thompson sampling, BUCB, and EBUCB with worst-case misspecified posteriors under different problem settings. Results are averaged over 10 runs with shaded standard errors.

$[0.7, 0.3]$, and use $\text{Beta}(1,1)$ as the prior distribution of mean reward for each action. At each time step $t$, the exact posterior distribution for each action is $\text{Beta}(1 + S_j(t), 1 + N_j(t) - S_j(t))$, where $S_j(t)$ is the empirical cumulative reward from action $j$ up to time $t$ and $N_j(t)$ is the number of draws from action $j$ up to time $t$.

### 4.1  Generally Misspecified Posteriors

Suppose the posterior distributions are misspecified to the following distributions:

$$(P1): (1-w) * \text{Beta}(1 + S_j(t), 1 + N_j(t) - S_j(t)) + w * \text{Beta}(\frac{1 + S_j(t)}{2}, \frac{1 + N_j(t) - S_j(t)}{2})$$

$$(P2): (1-w) * \text{Beta}(1 + S_j(t), 1 + N_j(t) - S_j(t)) + w * \text{Beta}(2(1 + S_j(t)), 2(1 + N_j(t) - S_j(t)))$$

where $w = 0.9, 0.8, 0.7$. Figure 1 presents the results of EBUCB and the baselines. Overall, EBUCB achieves consistently superior performance than the baselines, and it outperforms BUCB with considerable improvements. These results confirm the effectiveness of EBUCB across multiple settings. Moreover, EBUCB performs well without the horizon-dependent term (i.e., $c = 0$ in Corollary 3.8). This brings EBUCB practical advantages in real-world applications, as it does not require advanced knowledge of the horizon (i.e., *anytime*). A similar observation of BUCB was also noticed in [21].

## 4.2 Worst-Case Misspecified Posteriors

We consider the worst-case examples, Equations (13) and (14), presented in the proof of Theorems 3.12 and 3.13, where the posterior distributions are misspecified using one $\alpha$-divergence. The results of Thompson sampling, BUCB, and EBUCB are displayed in Figure 2. From these worst-case examples, we observe that: 1) Thompson sampling exhibits a linear regret after $t \geq 1$. As shown in Theorem 3.12, the linear coefficient (i.e., the slope) of the regret depends on the level $r$ that corresponds to the inference error. Specifically, the slope of the regret is increased along with the increased value of $r$, as illustrated in both Figure 2 and the proof of Theorem 3.12. 2) BUCB/EBUCB exhibits a linear regret with constant slope $\mu_1 - \mu_2$ after $t \geq T_0$, where $T_0$ is the time threshold introduced in Theorem 3.13 after which BUCB/EBUCB always chooses the sub-optimal action. The artificial choice of $r$ is to make $T_0 = 100, 200, 333$ where $\gamma_{T_0} = \frac{1}{r}$; See the proof of Theorem 3.13.

In summary, our experiments evidently demonstrate the superior performance of our proposed EBUCB on multi-armed bandit problems with generally misspecified posteriors. Our results also align closely with our theory that making one $\alpha$-divergence a small constant alone is insufficient to guarantee a sub-linear regret of Thompson sampling/BUCB/EBUCB. Hence, making two different $\alpha$-divergences bounded is necessary for the sub-linear regret upper bound.

## 5 Conclusions and Future Work

In this paper, we propose a general Bayesian bandit algorithm, Enhanced Bayesian Upper Confidence Bound (EBUCB), that achieves superior performance for Bernoulli bandit problems with approximate inference. We prove that, if the inference error measured by two different $\alpha$-divergences is less than a constant, EBUCB can achieve the optimal regret order $O(\log T)$. Additionally, we construct worse-case examples to show the necessity of bounding two different $\alpha$-divergences, which is further validated by our experiments. We consider the study of other problem settings as meaningful future research that could be built upon our current framework, e.g., extending to the general exponential family bandit problems by leveraging the techniques in [20]. We will also extend our current framework to contextual bandit problems and investigate the performance of contextual Bayesian bandit algorithms with approximate inference [36].

## Acknowledgments and Disclosure of Funding

This work has been supported in part by the National Science Foundation under grants CAREER CMMI-1834710 and IIS-1849280, and the Cheung-Kong Innovation Doctoral Fellowship. The authors thank the anonymous reviewers for their constructive comments which have helped greatly improve the quality of our paper.

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

# Appendices

We provide further results and discussions in this appendix. Section A presents detailed discussions on Assumption 3.3. Section B presents proofs for the results in Section 3.2 in the main paper. Section C presents proofs for the results in Section 3.3. Section D presents proofs for the results in Section 3.4. Section E presents additional experimental results.

## Appendix A    Remarks on Assumption 3.3

*Remark* A.1 (Constant Threshold). A constant threshold assumption on inference error appears in [34]. In this study, we adopt a similar assumption as [34], since no standard assumptions are in place due to the infancy of this field. In practice, keeping inference error below a constant threshold is not only feasible but also can be improved. As a concrete example, [30] showed that the inference error of an efficient Langevin MCMC algorithm decreases at the order of $O(1/\sqrt{N_j(t)})$. [48] also establish similar consistency results. On a high level, as more data are collected, the approximate and exact posteriors both become more concentrated, but they could concentrate to the same point of true mean rewards, as [30] showed. In addition to MCMC algorithms [4] which typically produce a consistent posterior, our assumption also fits other Bayesian inference algorithms, such as variational inference [8], that output approximate posteriors with a systematic computational bias.

In fact, the negative results in the presence of constant inference error shown by [34] motivate another research direction: A natural idea for obtaining positive results is to construct a "highly-effective" Bayesian inference approach with vanishing inference error as the number of samples increases, as studied in [30, 48]. However, our results demonstrate that the $O(\log T)$ regret upper bound is achievable even with non-vanishing inference error, without requiring specific "highly-effective" Bayesian inference approaches.

*Remark* A.2 ($\alpha$-divergence). Studies beyond KL divergence [38, 9], using $\alpha$-divergence [26, 19, 14, 49] or more generally $f$-divergence [46, 47], have appeared in recent research. These studies demonstrated the potential use of general $\alpha$-divergence and other types of divergence in practical model design. Moreover, from a practical perspective, the two $\alpha$-divergences between the estimated and the exact posterior distributions could be small, even if only one $\alpha$-divergence is used to construct the approximate distribution. This might be the reason for the superior performance of approximate Bayesian bandit algorithms that consider only the KL divergence in practice.

*Remark* A.3 (Independent Prior). In multi-armed bandit problems, the prior distribution of mean rewards $\Pi_0$ is typically chosen to be independent among each action $\Pi_{0,1}, ..., \Pi_{0,K}$, as this is the most reasonable and natural way without further information [1, 21, 20, 30]. In this case, the posteriors must also be independent among each action because of the Bayesian update (2). Therefore, our study focuses on the inference error on the distribution of each action in (3), which appears to be more realistic than the inference error on the joint distribution of all actions assumed in [34].

## Appendix B    Proofs of Results in Section 3.2

*Proof of Theorem 3.4.* Note that $R_1(\gamma) = R_2(\gamma + \delta_{\gamma,\epsilon})$ implies $R_2^{-1}(R_1(\gamma)) = \gamma + \delta_{\gamma,\epsilon}$ and $R_1^{-1}(R_2(\gamma + \delta_{\gamma,\epsilon})) = \gamma$. Obviously $R_2^{-1}(R_1(0)) = 0$ and $R_2^{-1}(R_1(1)) = 1$. On a high level, our proof technique is to split the $D_\alpha(P_1, P_2)$ into two parts using the quantile-based representation of $\alpha$-divergence in Lemma 3.5, and then use Jensen's inequality (since $\frac{1}{\alpha(\alpha-1)}x^{1-\alpha}$ is a convex function of $x \geq 0$ for any $\alpha \neq 0, 1$) to derive a bound for $\delta_{\gamma,\epsilon}$.

By Lemma 3.5 and Jensen's inequality,

$$D_\alpha(P_1, P_2)$$

$$=\frac{1}{\alpha(\alpha-1)}\int_0^1\left(\frac{d}{du}R_2^{-1}(R_1(u))\right)^{1-\alpha}du - \frac{1}{\alpha(\alpha-1)}$$

$$=\frac{1}{\alpha(\alpha-1)}\int_0^\gamma\left(\frac{d}{du}R_2^{-1}(R_1(u))\right)^{1-\alpha}du + \frac{1}{\alpha(\alpha-1)}\int_\gamma^1\log\left(\frac{d}{du}R_2^{-1}(R_1(u))\right)^{1-\alpha}du - \frac{1}{\alpha(\alpha-1)}$$

$$\geq\frac{\gamma}{\alpha(\alpha-1)}\left(\frac{1}{\gamma}\int_0^\gamma\frac{d}{du}R_2^{-1}(R_1(u))du\right)^{1-\alpha} + \frac{1-\gamma}{\alpha(\alpha-1)}\left(\frac{1}{1-\gamma}\int_\gamma^1\frac{d}{du}R_2^{-1}(R_1(u))du\right)^{1-\alpha} - \frac{1}{\alpha(\alpha-1)}$$

$$=\frac{\gamma}{\alpha(\alpha-1)}\left(\frac{R_2^{-1}(R_1(\gamma))-R_2^{-1}(R_1(0))}{\gamma}\right)^{1-\alpha} + \frac{1-\gamma}{\alpha(\alpha-1)}\left(\frac{R_2^{-1}(R_1(1))-R_2^{-1}(R_1(\gamma))}{1-\gamma}\right)^{1-\alpha} - \frac{1}{\alpha(\alpha-1)}$$

$$=\frac{\gamma}{\alpha(\alpha-1)}\left(\frac{\gamma+\delta_{\gamma,\epsilon}}{\gamma}\right)^{1-\alpha} + \frac{1-\gamma}{\alpha(\alpha-1)}\left(\frac{1-\gamma-\delta_{\gamma,\epsilon}}{1-\gamma}\right)^{1-\alpha} - \frac{1}{\alpha(\alpha-1)} \qquad (5)$$

First, we note that for any $\gamma \in (0,1)$ and any $\delta_{\gamma,\epsilon} \in [-\gamma, 1-\gamma]$, the above inequality is indeed achievable. To see this, consider

$$p_2(x) = \begin{cases} \frac{\gamma+\delta_{\gamma,\epsilon}}{\gamma}p_1(x) & \text{if } x_1 < R_1(\gamma) \\ \frac{1-\gamma-\delta_{\gamma,\epsilon}}{1-\gamma}p_1(x_1) & \text{if } x_1 > R_1(\gamma) \end{cases} \qquad (6)$$

Simple calculations give

$$\int p_2(x) = \frac{\gamma+\delta_{\gamma,\epsilon}}{\gamma}F_1(R_1(\gamma)) + \frac{1-\gamma-\delta_{\gamma,\epsilon}}{1-\gamma}(1-F_1(R_1(\gamma))) = \frac{\gamma+\delta_{\gamma,\epsilon}}{\gamma}\gamma + \frac{1-\gamma-\delta_{\gamma,\epsilon}}{1-\gamma}(1-\gamma) = 1$$

where $F_1$ is the cdf of $P_1$, and

$$D_\alpha(P_1, P_2)$$

$$=\frac{1}{\alpha(\alpha-1)}\left(\int\left(\frac{p_2(x)}{p_1(x)}\right)^{1-\alpha}p_1(x)dx - 1\right)$$

$$=\frac{1}{\alpha(\alpha-1)}\left(\left(\frac{\gamma+\delta_{\gamma,\epsilon}}{\gamma}\right)^{1-\alpha}F_1(R_1(\gamma)) + \left(\frac{1-\gamma-\delta_{\gamma,\epsilon}}{1-\gamma}\right)^{1-\alpha}(1-F_1(R_1(\gamma))) - 1\right)$$

$$=\frac{\gamma}{\alpha(\alpha-1)}\left(\frac{\gamma+\delta_{\gamma,\epsilon}}{\gamma}\right)^{1-\alpha} + \frac{1-\gamma}{\alpha(\alpha-1)}\left(\frac{1-\gamma-\delta_{\gamma,\epsilon}}{1-\gamma}\right)^{1-\alpha} - \frac{1}{\alpha(\alpha-1)}.$$

Therefore, the $P_1$ and $P_2$ constructed in Equation (6) show that the inequality (5) is indeed achievable.

Next, we consider the function

$$g(\delta) = \frac{\gamma}{\alpha(\alpha-1)}\left(\frac{\gamma+\delta}{\gamma}\right)^{1-\alpha} + \frac{1-\gamma}{\alpha(\alpha-1)}\left(\frac{1-\gamma-\delta}{1-\gamma}\right)^{1-\alpha} - \frac{1}{\alpha(\alpha-1)}.$$

Taking the derivative of $g(\delta)$ with respect to $\delta$, we obtain

$$g'(\delta) = -\frac{1}{\alpha}\left(\frac{\gamma+\delta}{\gamma}\right)^{-\alpha} + \frac{1}{\alpha}\left(\frac{1-\gamma-\delta}{1-\gamma}\right)^{-\alpha}.$$

$$g''(\delta) = \frac{1}{\gamma}\left(\frac{\gamma+\delta}{\gamma}\right)^{-\alpha-1} + \frac{1}{1-\gamma}\left(\frac{1-\gamma-\delta}{1-\gamma}\right)^{-\alpha-1} > 0.$$

It is easy to see that $g(\delta)$ is strictly decreasing when $\delta \in (-\gamma, 0)$ and strictly increasing when $\delta \in (0, 1-\gamma)$ with minimum $g(0) = 0$. This fact shows that $\epsilon \geq g(\delta)$ is equivalent to $\delta$ in an interval around 0, i.e., $\delta \in [\underline{\delta}, \overline{\delta}] \subset [-\gamma, 1-\gamma]$. More explicitly, we can obtain an explicit bound for $\underline{\delta}, \overline{\delta}$. In the following, we will show that in some cases, the bound for $\delta$ becomes too loose in the sense that $\underline{\delta}$ can be close to $-\gamma$ or $\overline{\delta}$ can be close to $1-\gamma$, in which cases we cannot obtain any useful information.

1) Suppose that $\alpha > 1$. When $\delta_{\gamma,\epsilon} > 0$, we have

$$\frac{\gamma}{\alpha(\alpha-1)}\left(\frac{\gamma+\delta_{\gamma,\epsilon}}{\gamma}\right)^{1-\alpha} \geq \frac{\gamma}{\alpha(\alpha-1)}\left(\frac{\gamma+1-\gamma}{\gamma}\right)^{1-\alpha} = \frac{\gamma^\alpha}{\alpha(\alpha-1)} \geq 0$$

as $0 < \delta_{\gamma,\epsilon} \leq 1 - \gamma$. Therefore $\epsilon \geq D_\alpha(P_1, P_2)$ implies that

$$\frac{1-\gamma}{\alpha(\alpha-1)}\left(\frac{1-\gamma-\delta_{\gamma,\epsilon}}{1-\gamma}\right)^{1-\alpha} \leq \epsilon + \frac{1}{\alpha(\alpha-1)}$$

which is equivalent to

$$\frac{1-\gamma-\delta_{\gamma,\epsilon}}{1-\gamma} \geq \left(\frac{\epsilon\alpha(\alpha-1)+1}{1-\gamma}\right)^{\frac{1}{1-\alpha}} .$$

Hence we have

$$\delta_{\gamma,\epsilon} \leq 1 - \gamma - (\epsilon\alpha(\alpha-1)+1)^{\frac{1}{1-\alpha}}(1-\gamma)^{\frac{\alpha}{\alpha-1}} .$$

**Remark.** Note that when $\alpha > 1$, the lower bound of $\delta_{\gamma,\epsilon}$ can be similarly derived but it becomes vicious. To see this, we notice that whenever

$$(\epsilon\alpha(\alpha-1)+\gamma)^{\frac{1}{1-\alpha}}\gamma^{\frac{\alpha}{\alpha-1}} - \gamma \leq \delta_{\gamma,\epsilon} < 0, \tag{7}$$

we have that

$$\frac{1-\gamma}{\alpha(\alpha-1)}\left(\frac{1-\gamma-\delta_{\gamma,\epsilon}}{1-\gamma}\right)^{1-\alpha} - \frac{1}{\alpha(\alpha-1)} \leq \frac{1-\gamma}{\alpha(\alpha-1)} - \frac{1}{\alpha(\alpha-1)} = \frac{-\gamma}{\alpha(\alpha-1)}$$

as $-\gamma \leq \delta_{\gamma,\epsilon} < 0$ and

$$\frac{\gamma}{\alpha(\alpha-1)}\left(\frac{\gamma+\delta_{\gamma,\epsilon}}{\gamma}\right)^{1-\alpha} \leq \frac{\gamma}{\alpha(\alpha-1)}\left(\frac{\gamma+(\epsilon\alpha(\alpha-1)+\gamma)^{\frac{1}{1-\alpha}}\gamma^{\frac{\alpha}{\alpha-1}}-\gamma}{\gamma}\right)^{1-\alpha} = \epsilon + \frac{\gamma}{\alpha(\alpha-1)}$$

which ensures $\epsilon \geq D_\alpha(P_1, P_2)$. However, the lower bound in Equation (7) is too loose:

$$\lim_{\epsilon\to+\infty}(\epsilon\alpha(\alpha-1)+\gamma)^{\frac{1}{1-\alpha}}\gamma^{\frac{\alpha}{\alpha-1}} - \gamma = -\gamma$$

which can be sufficiently close to $-\gamma$ if $\epsilon$ is sufficiently large.

b) Suppose that $\alpha < 0$. When $\delta_{\gamma,\epsilon} < 0$, we have

$$\frac{\gamma}{\alpha(\alpha-1)}\left(\frac{\gamma+\delta_{\gamma,\epsilon}}{\gamma}\right)^{1-\alpha} \geq \frac{\gamma}{\alpha(\alpha-1)}\left(\frac{\gamma-\gamma}{\gamma}\right)^{1-\alpha} = 0$$

as $-\gamma \leq \delta_{\gamma,\epsilon} < 0$. Therefore $\epsilon \geq D_\alpha(P_1, P_2)$ implies that

$$\frac{1-\gamma}{\alpha(\alpha-1)}\left(\frac{1-\gamma-\delta_{\gamma,\epsilon}}{1-\gamma}\right)^{1-\alpha} \leq \epsilon + \frac{1}{\alpha(\alpha-1)}$$

which is equivalent to

$$\frac{1-\gamma-\delta_{\gamma,\epsilon}}{1-\gamma} \leq \left(\frac{\epsilon\alpha(\alpha-1)+1}{1-\gamma}\right)^{\frac{1}{1-\alpha}} .$$

Hence we have

$$\delta_{\gamma,\epsilon} \geq 1 - \gamma - (\epsilon\alpha(\alpha-1)+1)^{\frac{1}{1-\alpha}}(1-\gamma)^{\frac{\alpha}{\alpha-1}} \tag{8}$$

**Remark.** Note that when $\alpha < 0$, the upper bound of $\delta_{\gamma,\epsilon}$ can be similarly derived but it becomes vicious. To see this, we notice that whenever

$$0 \leq \delta_{\gamma,\epsilon} < \min\{(\epsilon\alpha(\alpha-1)+\gamma)^{\frac{1}{1-\alpha}}\gamma^{\frac{\alpha}{\alpha-1}} - \gamma, 1-\gamma\} \tag{9}$$

we have that

$$\frac{1-\gamma}{\alpha(\alpha-1)}\left(\frac{1-\gamma-\delta_{\gamma,\epsilon}}{1-\gamma}\right)^{1-\alpha} - \frac{1}{\alpha(\alpha-1)} \leq \frac{1-\gamma}{\alpha(\alpha-1)} - \frac{1}{\alpha(\alpha-1)} = \frac{-\gamma}{\alpha(\alpha-1)}$$

as $-\gamma \le \delta_{\gamma,\epsilon} < 0$ and

$$\frac{\gamma}{\alpha(\alpha-1)}\left(\frac{\gamma+\delta_{\gamma,\epsilon}}{\gamma}\right)^{1-\alpha} \le \frac{\gamma}{\alpha(\alpha-1)}\left(\frac{\gamma+(\epsilon\alpha(\alpha-1)+\gamma)^{\frac{1}{1-\alpha}}\gamma^{\frac{\alpha}{\alpha-1}}-\gamma}{\gamma}\right)^{1-\alpha} = \epsilon+\frac{\gamma}{\alpha(\alpha-1)}$$

which ensures $\epsilon \ge D_\alpha(P_1, P_2)$. However, the lower bound in Equation (9) is too loose:

$$\lim_{\epsilon\to+\infty}(\epsilon\alpha(\alpha-1)+\gamma)^{\frac{1}{1-\alpha}}\gamma^{\frac{\alpha}{\alpha-1}}-\gamma = +\infty$$

so the lower bound in Equation (9) can be $1 - \gamma$ if $\epsilon$ is large.

c) Note that when $\alpha \in (0, 1)$, $\alpha(\alpha - 1) < 0$ and thus we have

$$\frac{\gamma}{\alpha(\alpha-1)}\left(\frac{\gamma+\delta_{\gamma,\epsilon}}{\gamma}\right)^{1-\alpha} \le \frac{\gamma}{\alpha(\alpha-1)}\left(\frac{\gamma-\gamma}{\gamma}\right)^{1-\alpha} = 0$$

and

$$\frac{1-\gamma}{\alpha(\alpha-1)}\left(\frac{1-\gamma-\delta_{\gamma,\epsilon}}{1-\gamma}\right)^{1-\alpha} \le \frac{1-\gamma}{\alpha(\alpha-1)}\left(\frac{1-\gamma-(1-\gamma)}{1-\gamma}\right)^{1-\alpha} = 0$$

as $-\gamma \le \delta_{\gamma,\epsilon} \le 1-\gamma$. Hence, as long as $\epsilon \ge \frac{-1}{\alpha(\alpha-1)}$, we always have

$$\frac{\gamma}{\alpha(\alpha-1)}\left(\frac{\gamma+\delta_{\gamma,\epsilon}}{\gamma}\right)^{1-\alpha} + \frac{1-\gamma}{\alpha(\alpha-1)}\left(\frac{1-\gamma-\delta_{\gamma,\epsilon}}{1-\gamma}\right)^{1-\alpha} - \frac{1}{\alpha(\alpha-1)} \le \epsilon$$

for any $\delta_{\gamma,\epsilon} \in [-\gamma, 1-\gamma]$. Moreover, as we discussed below Equation (5), for any $\delta_{\gamma,\epsilon} \in [-\gamma, 1-\gamma]$, there exist two distributions $P_1$ and $P_2$ such that

$$D_\alpha(P_1, P_2) = \frac{\gamma}{\alpha(\alpha-1)}\left(\frac{\gamma+\delta_{\gamma,\epsilon}}{\gamma}\right)^{1-\alpha} + \frac{1-\gamma}{\alpha(\alpha-1)}\left(\frac{1-\gamma-\delta_{\gamma,\epsilon}}{1-\gamma}\right)^{1-\alpha} - \frac{1}{\alpha(\alpha-1)},$$

which shows that $D_\alpha(P_1, P_2) \le \epsilon$ holds for such $P_1$ and $P_2$. This implies that the condition $D_\alpha(P_1, P_2) \le \epsilon$ cannot control the quantile shift between $P_1$ and $P_2$ in general when $\alpha \in (0, 1)$. $\quad\square$

*Proof of Lemma 3.5.* Let $F_1$ and $F_2$ be the cdfs of $P_1$ and $P_2$ respectively. Since $F_1$ and $F_2$ are absolute continuous and strictly increasing (as they have positive densities), we have that $F_i(R_i(u)) = u$ for $0 \le u \le 1$. Taking the derivative with respect to both sides of $F_i(R_i(u)) = u$, we obtain

$$R_i'(u)p_i(R_i(u)) = 1 \tag{10}$$

where $R_i'(u) := \frac{dR_i(u)}{du}$. Note that we have the following equality:

$$\begin{aligned}
\frac{d}{du}R_2^{-1}(R_1(u)) &= R_1'(u)\frac{dR_2^{-1}(v)}{dv}\Big|_{v=R_1(u)} \\
&= R_1'(u)\frac{1}{R_2'(R_2^{-1}(v))}\Big|_{v=R_1(u)} \\
&= R_1'(u)p_2(R_2(R_2^{-1}(v)))|_{v=R_1(u)} \quad \text{by Equation (10)} \\
&= R_1'(u)p_2(R_1(u)). \tag{11}
\end{aligned}$$

Using integration by substitution, for $\alpha$-divergence, we obtain that

$$
\begin{aligned}
D_\alpha(P_1, P_2) &= \frac{1}{\alpha(\alpha-1)} \left( \int p_1(x)^\alpha p_2(x)^{1-\alpha} dx - 1 \right) \\
&= \frac{1}{\alpha(\alpha-1)} \left( \int \left( \frac{p_1(x)}{p_2(x)} \right)^{\alpha-1} p_1(x) dx - 1 \right) \\
&= \frac{1}{\alpha(\alpha-1)} \left( \int_0^1 \left( \frac{p_1(R_1(u))}{p_2(R_1(u))} \right)^{\alpha-1} p_1(R_1(u)) dR_1(u) - 1 \right) \\
&= \frac{1}{\alpha(\alpha-1)} \left( \int_0^1 \left( \frac{1}{R_1'(u)p_2(R_1(u))} \right)^{\alpha-1} p_1(R_1(u)) R_1'(u) du - 1 \right) \\
&= \frac{1}{\alpha(\alpha-1)} \left( \int_0^1 \left( R_1'(u)p_2(R_1(u)) \right)^{1-\alpha} du - 1 \right) \quad \text{by Equation (10)} \\
&= \frac{1}{\alpha(\alpha-1)} \left( \int_0^1 \left( \frac{d}{du} R_2^{-1}(R_1(u)) \right)^{1-\alpha} du - 1 \right) \quad \text{by Equation (11).}
\end{aligned}
$$

Similarly, for KL divergence, we obtain that

$$
\begin{aligned}
KL(P_1, P_2) &= \int_0^1 \log \left( \frac{p_1(R_1(u))}{p_2(R_1(u))} \right) p_1(R_1(u)) dR_1(u) \\
&= \int_0^1 \log \left( \frac{1}{R_1'(u)p_2(R_1(u))} \right) p_1(R_1(u)) R_1'(u) du \\
&= -\int_0^1 \log \left( R_1'(u)p_2(R_1(u)) \right) du \quad \text{by Equation (10)} \\
&= -\int_0^1 \log \left( \frac{d}{du} R_2^{-1}(R_1(u)) \right) du \quad \text{by Equation (11).}
\end{aligned}
$$

$\square$

## Appendix C   Proofs of Results in Section 3.3

We first prove two useful lemmas.

**Lemma C.1.** *Under the same condition in Theorem 3.7, the quantiles of the approximate distributions* $qu_j(t)$ *chosen by the EBUCB algorithm satisfies the following bound:*

$$
\underline{u}_j(t) \le qu_j(t) \le \overline{u}_j(t)
$$

*where*

$$
\overline{u}_j(t) = \arg\max_{x > \frac{S_j(t)}{N_j(t)}} \left\{ d\left( \frac{S_j(t)}{N_j(t)}, x \right) \le \frac{\zeta\tilde{\alpha}_1 \log(t) + c\tilde{\alpha}_1 \log(\log T) - \log(M_{\epsilon,1})}{N_j(t)} \right\}.
$$

$$
\underline{u}_j(t) = \arg\max_{x > \frac{S_j(t)}{N_j(t)+1}} \left\{ d\left( \frac{S_j(t)}{N_j(t)+1}, x \right) \le \frac{\zeta\tilde{\alpha}_2 \log(t) + c\tilde{\alpha}_2 \log(\log T) - \log(M_{\epsilon,2}) - \log(N_j(t)+2)}{N_j(t)+1} \right\},
$$

*Proof of Lemma C.1.* Recall that $M_{\epsilon,1} = (\epsilon\alpha_1(\alpha_1-1)+1)^{\frac{1}{1-\alpha_1}} < 1$, $\tilde{\alpha}_1 = \frac{\alpha_1}{\alpha_1-1} > 0$, $M_{\epsilon,2} = (\epsilon\alpha_2(\alpha_2-1)+1)^{\frac{1}{1-\alpha_2}} > 1$, $\tilde{\alpha}_2 = \frac{\alpha_2}{\alpha_2-1} > 0$. First we notice that by Theorem 3.4 part a) (where $P_1$

corresponds to $Q_{t,j-1}$), we have

$$qu_j(t) = Qu(1 - \frac{1}{t^\zeta (\log T)^c}, Q_{t-1,j})$$

$$= Qu(1 - \frac{1}{t^\zeta (\log T)^c} + \delta_{1 - \frac{1}{t^\zeta (\log T)^c}, \epsilon}, \Pi_{t-1,j})$$

$$\leq Qu(1 - \frac{1}{t^\zeta (\log T)^c} + \frac{1}{t^\zeta (\log T)^c} - M_{\epsilon,1} \frac{1}{t^{\zeta \tilde{\alpha}_1} (\log T)^{c\tilde{\alpha}_1}}, \Pi_{t-1,j})$$

$$= Qu(1 - \frac{M_{\epsilon,1}}{t^{\zeta \tilde{\alpha}_1} (\log T)^{c\tilde{\alpha}_1}}, \Pi_{t-1,j})$$

since $D_{\alpha_1}(Q_{t,j-1}, \Pi_{t,j-1}) \leq \epsilon$ and we have use the fact that $Qu$ is non-decreasing. Now we apply the proof of Lemma 1 in [21], the tight bounds of the quantiles of the Beta distributions, to obtain

$$Qu(1 - \frac{M_{\epsilon,1}}{t^{\zeta \tilde{\alpha}_1} (\log T)^{c\tilde{\alpha}_1}}, \Pi_{t-1,j})$$

$$\leq \underset{x > \frac{S_j(t)}{N_j(t)}}{\arg \max} \left\{ d\left( \frac{S_j(t)}{N_j(t)}, x \right) \leq \frac{\log\left( \frac{1}{\frac{M_{\epsilon,1}}{t^{\zeta \tilde{\alpha}_1} (\log T)^{c\tilde{\alpha}_1}}} \right)}{N_j(t)} \right\}$$

$$\leq \underset{x > \frac{S_j(t)}{N_j(t)}}{\arg \max} \left\{ d\left( \frac{S_j(t)}{N_j(t)}, x \right) \leq \frac{\zeta \tilde{\alpha}_1 \log(t) + c\tilde{\alpha}_1 \log(\log T) - \log(M_{\epsilon,1})}{N_j(t)} \right\}.$$

$$= \overline{u}_j(t)$$

Similarly, by Theorem 3.4 part b) (where $P_1$ corresponds to $Q_{t,j-1}$), we have that

$$qu_j(t) = Qu(1 - \frac{1}{t^\zeta (\log T)^c}, Q_{t-1,j})$$

$$= Qu(1 - \frac{1}{t^\zeta (\log T)^c} + \delta_{1 - \frac{1}{t^\zeta (\log T)^c}, \epsilon}, \Pi_{t-1,j})$$

$$\geq Qu(1 - \frac{1}{t^\zeta (\log T)^c} + \frac{1}{t^\zeta (\log T)^c} - M_{\epsilon,2} \frac{1}{t^{\zeta \tilde{\alpha}_2} (\log T)^{c\tilde{\alpha}_2}}, \Pi_{t-1,j})$$

$$= Qu(1 - \frac{M_{\epsilon,2}}{t^{\zeta \tilde{\alpha}_2} (\log T)^{c\tilde{\alpha}_2}}, \Pi_{t-1,j})$$

since $D_{\alpha_2}(\Pi_{t,j-1}, Q_{t,j-1}) \leq \epsilon$ and we have use the fact that $Qu$ is non-decreasing. Now we apply the proof of Lemma 1 in [21], the tight bounds of the quantiles of the Beta distributions, to obtain

$$Qu(1 - \frac{M_{\epsilon,2}}{t^{\zeta \tilde{\alpha}_2} (\log T)^{c\tilde{\alpha}_2}}, \Pi_{t-1,j})$$

$$\geq \underset{x > \frac{S_j(t)}{N_j(t)+1}}{\arg \max} \left\{ d\left( \frac{S_j(t)}{N_j(t)+1}, x \right) \leq \frac{\log\left( \frac{1}{\frac{M_{\epsilon,2}}{t^{\zeta \tilde{\alpha}_2} (\log T)^{c\tilde{\alpha}_2}} (N_j(t)+2)} \right)}{N_j(t)+1} \right\}$$

$$\geq \underset{x > \frac{S_j(t)}{N_j(t)+1}}{\arg \max} \left\{ d\left( \frac{S_j(t)}{N_j(t)+1}, x \right) \leq \frac{\zeta \tilde{\alpha}_2 \log(t) + c\tilde{\alpha}_2 \log(\log T) - \log(M_{\epsilon,2}) - \log(N_j(t)+2)}{N_j(t)+1} \right\}$$

$$= \underline{u}_j(t)$$

Therefore, we conclude that

$$\underline{u}_j(t) \leq qu_j(t) \leq \overline{u}_j(t).$$

$\square$

Based on Lemma C.1, we can obtain a UCB-type decomposition of the number of draws of any sub-optimal action $j \geq 2$ as follows.

**Lemma C.2.** *Under the same condition in Theorem 3.7, we have that for any constant $\beta_T$,*

$$
\begin{aligned}
N_2(T) \leq &\sum_{t=1}^{T} \mathbb{1}\{\mu_1 - \beta_T > \underline{u}_1(t)\} \\
&+ \sum_{t=1}^{T} \mathbb{1}\{(\mu_1 - \beta_T \leq \overline{u}_2(t)) \cap (A_t = 2)\}.
\end{aligned}
\tag{12}
$$

*Proof of Lemma C.2.* We have that, by definition,

$$
\begin{aligned}
N_2(T) &= \sum_{t=1}^{T} \mathbb{1}\{A_t = 2\} \\
&= \sum_{t=1}^{T} \mathbb{1}\{(\mu_1 - \beta_T > q_1(t)) \cap (A_t = 2)\} + \sum_{t=1}^{T} \mathbb{1}\{(\mu_1 - \beta_T \leq q_1(t)) \cap (A_t = 2)\} \\
&\leq \sum_{t=1}^{T} \mathbb{1}\{\mu_1 - \beta_T > q_1(t)\} + \sum_{t=1}^{T} \mathbb{1}\{(\mu_1 - \beta_T \leq q_1(t)) \cap (A_t = 2)\} \\
&\leq \sum_{t=1}^{T} \mathbb{1}\{\mu_1 - \beta_T > \underline{u}_1(t)\} + \sum_{t=1}^{T} \mathbb{1}\{(\mu_1 - \beta_T \leq \overline{u}_2(t)) \cap (A_t = 2)\}
\end{aligned}
$$

where the last inequality follows from the fact that $q_1(t) \geq \underline{u}_1(t)$ and when $A_t = 2$, $q_1(t) \leq q_2(t) \leq \overline{u}_2(t)$. $\qquad\square$

Therefore, to obtain Theorem 3.7, it is sufficient to analyze the two terms in Lemma C.2.

*Proof of Theorem 3.7.* Without loss of generality, we let $j = 2$. (Note that we have assumed the action 1 is optimal.) By Lemma C.2, we only need to bound the two following two terms:

$$
D_1 := \sum_{t=1}^{T} \mathbb{1}\{\mu_1 - \beta_T > \underline{u}_1(t)\}, \quad D_2 := \sum_{t=1}^{T} \mathbb{1}\{(\mu_1 - \beta_T \leq \overline{u}_2(t)) \cap (A_t = 2)\}
$$

Let $\beta_T = \sqrt{\frac{1}{\log T}}$. We further split $D_1$ into two parts:

$$
D_{1,1} := \sum_{t=1}^{T} \mathbb{1}\{\mu_1 - \beta_T > \underline{u}_1(t), N_1(t) + 2 \leq (\log T)^2\},
$$

$$
D_{1,2} := \sum_{t=1}^{T} \mathbb{1}\{\mu_1 - \beta_T > \underline{u}_1(t), N_1(t) + 2 \geq (\log T)^2\}.
$$

**Step 1**: Consider $D_{1,1}$.

Note that

$$
\frac{\zeta\tilde{\alpha}_2 \log(t) + c\tilde{\alpha}_2 \log(\log T) - \log(M_{\epsilon,2}) - \log(N_1(t) + 2)}{N_1(t) + 1} \geq \frac{\zeta\tilde{\alpha}_2 \log(t) + (c\tilde{\alpha}_2 - 2) \log(\log T) - \log(M_{\epsilon,2})}{N_1(t) + 1}
$$

in $\underline{u}_1(t)$ when $N_1(t) + 2 \leq (\log T)^2$. Hence we have that

$$
\underline{u}_1(t) \geq \underset{x > \frac{S_1(t)}{N_1(t)+1}}{\arg\max} \left\{ d\left(\frac{S_1(t)}{N_1(t) + 1}, x\right) \leq \frac{\zeta\tilde{\alpha}_2 \log(t) + (c\tilde{\alpha}_2 - 2) \log(\log T) - \log(M_{\epsilon,2})}{N_1(t) + 1} \right\} := \tilde{\underline{u}}_1(t)
$$

when $N_1(t) + 2 \leq (\log T)^2$. This shows that

$$
D_{1,1} \leq \sum_{t=1}^{T} \mathbb{1}\{\mu_1 > \tilde{\underline{u}}_1(t)\}
$$

Similarly to the proof in [21], with a straightforward adaptation of the proof of theorem 10 in [15], we obtain the following self-normalized inequality

**Lemma C.3.**
$$\mathbb{P}(\mu_1 > \underline{\tilde{u}}_1(t)) \leq (\bar{\delta} \log(t) + 1) \exp(-\bar{\delta} + 1)$$

*where*
$$\bar{\delta} = \zeta \tilde{\alpha}_2 \log(t) + (c\tilde{\alpha}_2 - 2) \log(\log T) - \log(M_{\epsilon,2}).$$

Lemma C.3 leads to the following upper bound of $D_{1,1}$:

$$\mathbb{E}[D_{1,1}] \leq \sum_{t=1}^{T} \mathbb{P}(\mu_1 > \underline{\tilde{u}}_1(t))$$

$$\leq 1 + \sum_{t=2}^{T} \Big( \big( (\zeta \tilde{\alpha}_2 \log(t) + (c\tilde{\alpha}_2 - 2) \log(\log T) - \log(M_{\epsilon,2})) \log(t) + 1 \Big) \frac{M_{\epsilon,2} e}{t^{\zeta \tilde{\alpha}_2} (\log T)^{c\tilde{\alpha}_2 - 4}}$$

$$\leq 1 + \sum_{t=2}^{T} \Big( \zeta \tilde{\alpha}_2 + c\tilde{\alpha}_2 - 2 + 1 \Big) \frac{M_{\epsilon,2} e}{t^{\zeta \tilde{\alpha}_2} (\log T)^{c\tilde{\alpha}_2 - 4}}$$

$$\leq 1 + \Big( \zeta \tilde{\alpha}_2 + c\tilde{\alpha}_2 \Big) \frac{M_{\epsilon,2} e}{(\log T)^{c\tilde{\alpha}_2 - 4}} \sum_{t=2}^{T} \int_{t-1}^{t} \frac{1}{x^{\zeta \tilde{\alpha}_2}} dx$$

$$\leq 1 + \Big( \zeta \tilde{\alpha}_2 + c\tilde{\alpha}_2 \Big) \frac{M_{\epsilon,2} e}{(\log T)^{c\tilde{\alpha}_2 - 4}} \int_{T}^{1} \frac{1}{x^{\zeta \tilde{\alpha}_2}} dx$$

$$\leq \begin{cases} 1 + \Big( \zeta \tilde{\alpha}_2 + c\tilde{\alpha}_2 \Big) \frac{M_{\epsilon,2} e}{(\log T)^{c\tilde{\alpha}_2 - 4}} \frac{T^{1-\zeta\tilde{\alpha}_2} - 1}{1 - \zeta\tilde{\alpha}_2} \leq 1 + \frac{\Big( \zeta\tilde{\alpha}_2 + c\tilde{\alpha}_2 \Big) M_{\epsilon,2} e T^{1-\zeta\tilde{\alpha}_2}}{1 - \zeta\tilde{\alpha}_2} & \text{if } 0 < \zeta\tilde{\alpha}_2 < 1 \\ 1 + \Big( \zeta \tilde{\alpha}_2 + c\tilde{\alpha}_2 \Big) \frac{M_{\epsilon,2} e}{(\log T)^{c\tilde{\alpha}_2 - 4}} \log(T) \leq 1 + \Big( \zeta\tilde{\alpha}_2 + c\tilde{\alpha}_2 \Big) M_{\epsilon,2} e & \text{if } \zeta\tilde{\alpha}_2 \geq 1 \end{cases}$$

for $c\tilde{\alpha}_2 \geq 5$.

**Step 2**: Consider $D_{1,2}$. Note that in this term, the optimal action 1 has been sufficiently drawn to be well estimated, so we can use a loose bound

$$\mathbb{E}[D_{1,1}] \leq \sum_{t=1}^{T} \mathbb{1}\{\mu_1 - \beta_T > \frac{S_1(t)}{N_1(t) + 1}, N_1(t) + 2 \geq (\log T)^2\}$$

This right-hand side only depends on the draws from action 1 and has been studied in Theorem 1 in [21], so we apply their results:

$$\mathbb{E}[D_{1,2}] \leq \frac{1}{T-1}.$$

**Step 3**: Consider $D_2$. Using the same technique as in lemma 7 in [15], $D_2$ is bounded by

$$D_2 \leq \sum_{s=1}^{T} \mathbb{1}\{sd^+(\hat{\mu}_2(t), \mu_1 - \beta_T) \leq \zeta \tilde{\alpha}_1 \log(T) + c\tilde{\alpha}_1 \log(\log T) - \log(M_{\epsilon,1})\}$$

For $\xi > 0$, we let

$$K_{T,\epsilon} = \frac{(1 + \xi)(\zeta \tilde{\alpha}_1 \log(T) + c\tilde{\alpha}_1 \log(\log T) - \log(M_{\epsilon,1}))}{d(\mu_2, \mu_1)}.$$

Then $D_2$ can be rewritten as

$$D_2 \leq \sum_{s=1}^{\lfloor K_{T,\epsilon} \rfloor} 1 + \sum_{s=\lfloor K_{T,\epsilon} \rfloor + 1}^{T} \mathbb{1}\{K_{T,\epsilon} d^+(\hat{\mu}_2(t), \mu_1 - \beta_T) \leq \zeta \tilde{\alpha}_1 \log(T) + c\tilde{\alpha}_1 \log(\log T) - \log(M_{\epsilon,1})\}$$

$$= \lfloor K_{T,\epsilon} \rfloor + \sum_{s=\lfloor K_{T,\epsilon} \rfloor + 1}^{T} \mathbb{1}\{d^+(\hat{\mu}_2(t), \mu_1 - \beta_T) \leq \frac{d(\mu_2, \mu_1)}{1 + \xi}\}$$

$$\leq K_{T,\epsilon} + \sum_{s=\lfloor K_{T,\epsilon} \rfloor + 1}^{T} \mathbb{1}\{d^+(\hat{\mu}_2(t), \mu_1) \leq \frac{d(\mu_2, \mu_1)}{1 + \xi} + \beta_T \frac{2}{\mu_1(2 - \mu_1)}\}$$

where the last inequality follows from the same technique in the proof of Theorem 1 in [21], by noting that the function $g(q) = d^+(\hat{\mu}_2(s), q)$ is convex and differentiable and $g'(q) = \frac{q - \hat{\mu}_2(s)}{q(1-q)} \mathbb{1}\{(q > \hat{\mu}_2(s)\}$. Hence, for $T \geq \exp\left(\left(\frac{2(1+\xi)(1+\xi/2)}{\xi\mu_1(1-\mu_1)d(\mu_2,\mu_1)}\right)^2\right)$, we obtain $\frac{d(\mu_2,\mu_1)}{1+\xi} + \beta_T \frac{2}{\mu_1(1-\mu)} \leq \frac{d(\mu_2,\mu_1)}{1+\xi/2}$. Following the proof of Theorem 1 in [21] (as well as [15]), we obtain

$$\mathbb{E}[D_2] \leq K_{T,\epsilon} + \sum_{s=\lfloor K_{T,\epsilon} \rfloor + 1}^{T} \mathbb{P}\left(d^+(\hat{\mu}_2(t), \mu_1) \leq \frac{d(\mu_2,\mu_1)}{1+\xi/2}\right)$$

$$\leq K_{T,\epsilon} + \frac{(1+\xi/2)^2}{\xi^2(\min\{\mu_2(1-\mu_2), \mu_1(1-\mu_1)\})^2}$$

**Step 4**: Combing the above results, we obtain that if $0 < \zeta\tilde{\alpha}_2 < 1$,

$$\mathbb{E}[N_2(T)] \leq \mathbb{E}[D_{1,1}] + \mathbb{E}[D_{1,2}] + \mathbb{E}[D_2]$$

$$\leq 1 + \frac{\left(\zeta\tilde{\alpha}_2 + c\tilde{\alpha}_2\right)M_{\epsilon,2}eT^{1-\zeta\tilde{\alpha}_2}}{1 - \zeta\tilde{\alpha}_2} + \frac{(1+\xi)(\zeta\tilde{\alpha}_1\log(T) + c\tilde{\alpha}_1\log(\log T) - \log(M_{\epsilon,1})}{d(\mu_2,\mu_1)}$$

$$+ \frac{(1+\xi/2)^2}{\xi^2(\min\{\mu_2(1-\mu_2), \mu_1(1-\mu_1)\})^2}$$

$$\leq \frac{\left(\zeta\tilde{\alpha}_2 + c\tilde{\alpha}_2\right)M_{\epsilon,2}eT^{1-\zeta\tilde{\alpha}_2}}{1 - \zeta\tilde{\alpha}_2} + o(T^{1-\zeta\tilde{\alpha}_2}).$$

and if $\zeta\tilde{\alpha}_2 \geq 1$,

$$\mathbb{E}[N_2(T)] \leq \mathbb{E}[D_{1,1}] + \mathbb{E}[D_{1,2}] + \mathbb{E}[D_2]$$

$$\leq 1 + \left(\zeta\tilde{\alpha}_2 + c\tilde{\alpha}_2\right)M_{\epsilon,2}e + \frac{(1+\xi)(\zeta\tilde{\alpha}_1\log(T) + c\tilde{\alpha}_1\log(\log T) - \log(M_{\epsilon,1})}{d(\mu_2,\mu_1)}$$

$$+ \frac{(1+\xi/2)^2}{\xi^2(\min\{\mu_2(1-\mu_2), \mu_1(1-\mu_1)\})^2}$$

$$\leq \frac{(1+\xi)\zeta\tilde{\alpha}_1}{d(\mu_2,\mu_1)}\log(T) + o(\log T).$$

$\square$

Note that the final step (Step 4) in the above proof provides an exact finite-time regret bound that holds for any time horizon $T$. It explicitly expresses the $o(\cdot)$ term in Theorem 3.7. The error terms $M_{\epsilon,1}^{-1}$ and $M_{\epsilon,2}$ depending on $\epsilon$ appear in this regret bound explicitly. Obviously, $M_{\epsilon,1}^{-1}$ and $M_{\epsilon,2}$ in the bound increase as $\epsilon$ increases.

In general, our above derivations depend on bounds for specific distributions (Beta posterior distributions with inference errors in our setting). It is a direction to generalize these results to more general bandit problems or more general families of distributions. For instance, [20] extends the setting of Beta posterior distribution to the exponential family that includes Gaussian (without approximate inference). Combining [20] with our techniques in Sections 3.2 and 3.3 may lead to analyzing the exponential family with approximate inference. This, however, requires some additional careful technical derivation beyond our current bounds in Bernoulli with approximate inference, which is a future research direction.

## Appendix D  Proofs of Results in Section 3.4

*Proof of Theorem 3.12.* We can explicitly construct such a distribution $Q_t$ as follows:

$$q_{t,2}(x_2) = \pi_{t,2}(x_2)$$

$$q_{t,1}(x_1) = \begin{cases} \frac{1 - \frac{1}{r}(1 - F_{t,1}(b_t))}{F_{t,1}(b_t)}\pi_{t,1}(x_1) & \text{if } 0 < x_1 < b_t \\ \frac{1}{r}\pi_{t,1}(x_1) & \text{if } b_t < x_1 < 1 \end{cases} \tag{13}$$

where $F_{t,1}$ is the cumulative distribution function (cdf) of $\Pi_{t,1}$ and $r > 1$, $b_t \in (0,1)$ will be specified later.

First, note that by setting

$$q_{t,2} = \pi_{t,2},$$

we have $D_\alpha(\Pi_{t,2}, Q_{t,2}) = 0$ and $q_{t,2} = \pi_{t,2}$ with the same support $[0,1]$, satisfying Assumptions 3.6 and 3.11 on action $j = 2$.

We set $b_t = Qu(\frac{1}{2}, \Pi_{t,2}) \in (0,1)$, the $\frac{1}{2}$-quantile of the distribution $\Pi_{t,2}$ (or equivalently, $Q_{t,2}$). Let $F_{t,1}$ be the cdf of $\Pi_{t,1}$. We have $F_{t,1}(b_t) \in (0,1)$. For $r > 1$, we set

$$q_{t,1}(x_1) = \begin{cases} \frac{1 - \frac{1}{r}(1 - F_{t,1}(b_t))}{F_{t,1}(b_t)} \pi_{t,1}(x_1) & \text{if } 0 < x_1 < b_t \\ \frac{1}{r}\pi_{t,1}(x_1) & \text{if } b_t < x_1 < 1 \end{cases}$$

Step 1: We show that $q_{t,1}(x_1)$ is indeed a density satisfying Assumption 3.6 on action $j = 1$. First of all, it is obvious that $q_{t,1} > 0$ on $(0,1)$ as $\pi_{t,1} > 0$ on $(0,1)$. Moreover

$$\int_0^1 q_{t,1}(x_1)dx_1 = \int_0^{b_t} q_{t,1}(x_1)dx_1 + \int_{b_t}^1 q_{t,1}(x_1)dx_1$$

$$= \int_0^{b_t} \frac{1 - \frac{1}{r}(1 - F_{t,1}(b_t))}{F_{t,1}(b_t)}\pi_{t,1}(x_1)dx_1 + \int_{b_t}^1 \frac{1}{r}\pi_{t,1}(x_1)dx_1$$

$$= \frac{1 - \frac{1}{r}(1 - F_{t,1}(b_t))}{F_{t,1}(b_t)}F_{t,1}(b_t) + \frac{1}{r}(1 - F_{t,1}(b_t))$$

$$= 1.$$

Step 2: We show that there exists an $r > 1$ (independent of $t$) such that $q_{t,1}$ satisfies Assumption 3.11 on action $j = 1$.

We have that when $\alpha < 0$ or $0 < \alpha < 1$:

$$D_\alpha(Q_{t,1}, \Pi_{t,1}) = \frac{1}{\alpha(\alpha - 1)}\left(\int_0^{b_t} \pi_{t,1}(x_1)\left(\frac{\pi_{t,1}(x_1)}{q_{t,1}(x_1)}\right)^{-\alpha}dx_1 + \int_{b_t}^1 \pi_{t,1}(x_1)\left(\frac{\pi_{t,1}(x_1)}{q_{t,1}(x_1)}\right)^{-\alpha}dx_1 - 1\right)$$

$$= \frac{1}{\alpha(\alpha - 1)}\left(\int_0^{b_t} \pi_{t,1}(x_1)\left(\frac{F_{t,1}(b_t)}{1 - \frac{1}{r}(1 - F_{t,1}(b_t))}\right)^{-\alpha}dx_1 + \int_{b_t}^1 \pi_{t,1}(x_1)r^{-\alpha}dx_1 - 1\right)$$

$$= \frac{1}{\alpha(\alpha - 1)}\left(\left(\frac{F_{t,1}(b_t)}{1 - \frac{1}{r}(1 - F_{t,1}(b_t))}\right)^{-\alpha}F_{t,1}(b_t) + r^{-\alpha}(1 - F_{t,1}(b_t)) - 1\right)$$

We note that

$$\frac{F_{t,1}(b_t)}{1 - \frac{1}{r}(1 - F_{t,1}(b_t))} \leq \frac{r - 1 + F_{t,1}(b_t)}{1 - \frac{1}{r}(1 - F_{t,1}(b_t))} = r$$

as $r > 1$. Hence we have

$$D_\alpha(Q_{t,1}, \Pi_{t,1}) \leq \frac{1}{\alpha(\alpha - 1)}\left(r^{-\alpha}F_{t,1}(b_t) + r^{-\alpha}(1 - F_{t,1}(b_t)) - 1\right) = \frac{1}{\alpha(\alpha - 1)}\left(r^{-\alpha} - 1\right).$$

Then for $1 < r < (\epsilon\alpha(\alpha - 1) + 1)^{-\frac{1}{\alpha}}$ (only if $\epsilon\alpha(\alpha - 1) + 1 > 0$, otherwise we put $+\infty$ as the upper bound on $r$), we have that

$$D_\alpha(Q_{t,1}, \Pi_{t,1}) \leq \epsilon.$$

When $\alpha = 0$:

$$D_0(Q_{t,1}, \Pi_{t,1}) = KL(\Pi_{t,1}, Q_{t,1})$$

$$= \int_0^{b_t} \pi_{t,1}(x_1)\log\left(\frac{\pi_{t,1}(x_1)}{q_{t,1}(x_1)}\right)dx_1 + \int_{b_t}^1 \pi_{t,1}(x_1)\log\left(\frac{\pi_{t,1}(x_1)}{q_{t,1}(x_1)}\right)dx_1$$

$$= \int_0^{b_t} \pi_{t,1}(x_1)\log\left(\frac{F_{t,1}(b_t)}{1 - \frac{1}{r}(1 - F_{t,1}(b_t))}\right)dx_1 + \int_{b_t}^1 \pi_{t,1}(x_1)\log(r)dx_1$$

$$= \log\left(\frac{F_{t,1}(b_t)}{1 - \frac{1}{r}(1 - F_{t,1}(b_t))}\right)F_{t,1}(b_t) + \log(r)(1 - F_{t,1}(b_t))$$

We note that

$$\frac{F_{t,1}(b_t)}{1 - \frac{1}{r}(1 - F_{t,1}(b_t))} \leq \frac{r - 1 + F_{t,1}(b_t)}{1 - \frac{1}{r}(1 - F_{t,1}(b_t))} = r$$

as $r > 1$. Hence we have

$$KL(\Pi_{t,1}, Q_{t,1}) \leq \log(r) F_{t,1}(b_t) + \log(r)(1 - F_{t,1}(b_t)) = \log(r).$$

Then for $1 < r < e^\epsilon$, we have that

$$D_0(Q_{t,1}, \Pi_{t,1}) = KL(\Pi_{t,1}, Q_{t,1}) \leq \log(r) \leq \log(e^\epsilon) = \epsilon.$$

Step 3: We show that the probability of sampling from $Q_{t-1}$ choosing action 2 is greater than a positive constant $\frac{1}{2}(1 - \frac{1}{r})$, which thus leads to a linear regret.

In fact, the probability of sampling from $Q_{t-1}$ choosing action 2 is given by $\mathbb{P}_{Q_{t-1}}(x_2 \geq x_1)$. Therefore we have that

$$\mathbb{P}_{Q_{t-1}}(x_2 \geq x_1) \geq \mathbb{P}_{Q_{t-1}}(x_2 \geq b_{t-1} \geq x_1) = \mathbb{P}_{Q_{t-1,2}}(x_2 \geq b_{t-1})\mathbb{P}_{Q_{t-1,1}}(x_1 \leq b_{t-1})$$

since $Q_{t-1,1}$ and $Q_{t-1,2}$ are independent.

$$\mathbb{P}_{Q_{t-1,2}}(x_2 \geq b_{t-1}) = \frac{1}{2}$$

since $b_{t-1}$ is the $\frac{1}{2}$-quantile of the distribution $\Pi_{t-1,2}$ and $p_{t-1,2} > 0$ on $(0, 1)$.

$$\mathbb{P}_{Q_{t-1,1}}(x_1 \leq b_{t-1}) = 1 - \mathbb{P}_{Q_{t-1,1}}(x_1 \geq b_{t-1}) = 1 - \frac{1}{r}(1 - F_{t-1,1}(b_{t-1})) \geq 1 - \frac{1}{r}$$

by our construction of $Q_{t-1,1}$. Therefore we have that

$$\mathbb{P}_{Q_{t-1}}(x_2 \geq x_1) \geq \frac{1}{2}(1 - \frac{1}{r}) > 0.$$

We conclude that the lower bound of the average expected regret is given by

$$R(T, \mathcal{A}) = \sum_{j=1}^{2}(\mu_1 - \mu_j)\mathbb{E}[N_j(t)] \geq (\mu_1 - \mu_2)\frac{T}{2}(1 - \frac{1}{r}) = \Omega(T)$$

leading to a linear regret. □

*Proof of Theorem 3.13.* We can explicitly construct such a distribution $Q_t$ as follows:

$$q_{t,1}(x_1) = \pi_{t,1}(x_1)$$

$$q_{t,2}(x_2) = \begin{cases} \frac{1}{r}\pi_{t,2}(x_2) & \text{if } 0 < x_1 < b_t \\ \frac{1 - \frac{1}{r}F_{t,2}(b_t)}{1 - F_{t,2}(b_t)}\pi_{t,2}(x_2) & \text{if } b_t < x_1 < 1 \end{cases} \tag{14}$$

where $F_{t,2}$ is the cdf of $\Pi_{t,2}$ and $r > 1$, $b_t \in (0, 1)$ will be specified later.

First, note that by setting

$$q_{t,1} = \pi_{t,1},$$

we have $D_\alpha(\Pi_{t,1}, Q_{t,1}) = 0$ and $q_{t,1} = \pi_{t,1}$ with the same support $[0, 1]$, satisfying Assumptions 3.6 and 3.11 on action $j = 1$.

We set $b_t = Qu(\gamma_{t+1}, \Pi_{t,1}) \in (0, 1)$, the $\gamma_{t+1}$-quantile of the distribution $\Pi_{t,1}$ (or equivalently, $Q_{t,1}$). Let $F_{t,2}$ be the cdf of $\Pi_{t,2}$. We have $F_{t,2}(b_t) \in (0, 1)$. For $r > 1$, we set

$$q_{t,2}(x_2) = \begin{cases} \frac{1}{r}\pi_{t,2}(x_2) & \text{if } 0 < x_2 < b_t \\ \frac{1 - \frac{1}{r}F_{t,2}(b_t)}{1 - F_{t,2}(b_t)}\pi_{t,2}(x_2) & \text{if } b_t < x_2 < 1 \end{cases}$$

Step 1: We show that $q_{t,2}(x_2)$ is indeed a density satisfying Assumption 3.6 on action $j = 2$. First of all, it is obvious that $q_{t,2} > 0$ on $(0,1)$ as $\pi_{t,2} > 0$ on $(0,1)$. Moreover

$$
\int_0^1 q_{t,2}(x_2)dx_2 = \int_0^{b_t} q_{t,2}(x_2)dx_2 + \int_{b_t}^1 q_{t,2}(x_2)dx_2
$$

$$
= \int_0^{b_t} \frac{1}{r}\pi_{t,2}(x_2)dx_2 + \int_{b_t}^1 \frac{1 - \frac{1}{r}F_{t,2}(b_t)}{1 - F_{t,2}(b_t)}\pi_{t,2}(x_2)dx_2
$$

$$
= \frac{1}{r}F_{t,2}(b_t) + \frac{1 - \frac{1}{r}F_{t,2}(b_t)}{1 - F_{t,2}(b_t)}(1 - F_{t,2}(b_t))
$$

$$
= 1.
$$

Step 2: We show that there exists an $r > 1$ (independent of $t$) such that $q_{t,2}$ satisfies Assumption 3.11 on action $j = 2$.

We have that when $\alpha < 0$ or $0 < \alpha < 1$:

$$
D_\alpha(Q_{t,2}, \Pi_{t,2}) = \frac{1}{\alpha(\alpha-1)}\left(\int_0^{b_t} \pi_{t,2}(x_2)\left(\frac{\pi_{t,2}(x_2)}{q_{t,2}(x_2)}\right)^{-\alpha}dx_2 + \int_{b_t}^1 \pi_{t,2}(x_2)\left(\frac{\pi_{t,2}(x_2)}{q_{t,2}(x_2)}\right)^{-\alpha}dx_2 - 1\right)
$$

$$
= \frac{1}{\alpha(\alpha-1)}\left(\int_0^{b_t} \pi_{t,2}(x_2)r^{-\alpha}dx_2 + \int_{b_t}^1 \pi_{t,2}(x_2)\left(\frac{1 - F_{t,2}(b_t)}{1 - \frac{1}{r}F_{t,2}(b_t)}\right)^{-\alpha}dx_2 - 1\right)
$$

$$
= \frac{1}{\alpha(\alpha-1)}\left(r^{-\alpha}F_{t,2}(b_t) + \left(\frac{1 - F_{t,2}(b_t)}{1 - \frac{1}{r}F_{t,2}(b_t)}\right)^{-\alpha}(1 - F_{t,2}(b_t)) - 1\right)
$$

We note that

$$
\frac{1 - F_{t,2}(b_t)}{1 - \frac{1}{r}F_{t,2}(b_t)} \leq \frac{r - F_{t,2}(b_t)}{1 - \frac{1}{r}F_{t,2}(b_t)} = r
$$

as $r > 1$. Hence we have

$$
D_\alpha(\Pi_{t,2}, Q_{t,2}) \leq \frac{1}{\alpha(\alpha-1)}\left(r^{-\alpha}F_{t,2}(b_t) + r^{-\alpha}(1 - F_{t,2}(b_t)) - 1\right) = \frac{1}{\alpha(\alpha-1)}\left(r^{-\alpha} - 1\right).
$$

Then for $1 < r < (\epsilon\alpha(\alpha-1) + 1)^{-\frac{1}{\alpha}}$ (only if $\epsilon\alpha(\alpha-1) + 1 > 0$, otherwise we put $+\infty$ as the upper bound on $r$), we have that

$$
D_\alpha(Q_{t,2}, \Pi_{t,2}) \leq \epsilon.
$$

When $\alpha = 0$,

$$
D_0(Q_{t,2}, \Pi_{t,2}) = KL(\Pi_{t,2}, Q_{t,2}) = \int_0^{b_t} \pi_{t,2}(x_2)\log\left(\frac{\pi_{t,2}(x_2)}{q_{t,2}(x_2)}\right)dx_2 + \int_{b_t}^1 \pi_{t,2}(x_2)\log\left(\frac{\pi_{t,2}(x_2)}{q_{t,2}(x_2)}\right)dx_2
$$

$$
= \int_0^{b_t} \pi_{t,2}(x_2)\log(r)\,dx_2 + \int_{b_t}^1 \pi_{t,2}(x_2)\log\left(\frac{1 - F_{t,2}(b_t)}{1 - \frac{1}{r}F_{t,2}(b_t)}\right)dx_2
$$

$$
= \log(r)F_{t,2}(b_t) + \log\left(\frac{1 - F_{t,2}(b_t)}{1 - \frac{1}{r}F_{t,2}(b_t)}\right)(1 - F_{t,2}(b_t))
$$

We note that

$$
\frac{1 - F_{t,2}(b_t)}{1 - \frac{1}{r}F_{t,2}(b_t)} \leq \frac{r - F_{t,2}(b_t)}{1 - \frac{1}{r}F_{t,2}(b_t)} = r
$$

as $r > 1$. Hence we have

$$
KL(\Pi_{t,2}, Q_{t,2}) \leq \log(r)F_{t,2}(b_t) + \log(r)(1 - F_{t,2}(b_t)) = \log(r).
$$

Then for $1 < r < e^\epsilon$, we have that

$$
D_0(Q_{t,2}, \Pi_{t,2}) = KL(\Pi_{t,2}, Q_{t,2}) \leq \log(r) = \log(e^\epsilon) = \epsilon.
$$

Therefore, we conclude that there exists an $r > 1$ (independent of $t$) such that $q_{t,2}$ satisfies Assumption 3.11 on action $j = 2$. Take this $r > 1$ and notice that since $\gamma_t \to 1$ as $t \to +\infty$, there must exists a $T_0 > 0$ such that for any $t \geq T_0$, we have that $\gamma_t > \frac{1}{r}$.

Step 3: We show that the EBUCB algorithm always chooses action 2 when $t \geq T_0$, which thus leads to a linear regret.

We note that when $t \geq T_0$, by definition,

$$\mathbb{P}_{Q_{t-1,1}}(x_1 \leq b_{t-1}) = \mathbb{P}_{\Pi_{t-1,1}}(x_1 \leq b_{t-1}) = \gamma_t,$$

$$\mathbb{P}_{Q_{t-1,2}}(x_2 \leq b_{t-1}) = \frac{1}{r}F_{t-1,2}(b_{t-1}) \leq \frac{1}{r} < \gamma_t,$$

which implies that

$$Qu(\gamma_t, Q_{t-1,1}) = b_{t-1} < Qu(\gamma_t, Q_{t-1,2})$$

Therefore after time step $t \geq T_0$, the EBUCB algorithm will always choose the action 2. We conclude that the lower bound of the average expected regret is given by

$$R(T, \mathcal{A}) = \sum_{j=1}^{2}(\mu_1 - \mu_j)\mathbb{E}\left[N_j(t)\right] \geq (\mu_1 - \mu_2)(T - T_0) = \Omega(T)$$

leading to a linear regret. □

## Appendix E   Additional Experiments

In this section, we present additional experimental results. We enrich our experiments by studying an increasing number of arms as well as multiple new problem instances with different inference errors. These results further support our findings in Section 4 that EBUCB without the horizon-dependent term (i.e., $c = 0$) performs the best.

Suppose the posterior distributions are misspecified to the following distributions:

$$(1 - w) * \text{Beta}(1 + S_j(t), 1 + N_j(t) - S_j(t)) + w * \text{Beta}(\Gamma(1 + S_j(t)), \Gamma(1 + N_j(t) - S_j(t)))$$

where $j \in [K]$. We conduct two experiments: 1) The Bernoulli multi-armed bandit problem has $K$ actions with the following mean rewards:

$K = 2$: mean rewards = [0.7, 0.3]

$K = 4$: mean rewards = [0.9, 0.7, 0.5, 0.3]

$K = 8$: mean rewards = [0.9, 0.8, 0.7, 0.6, 0.5, 0.4, 0.3, 0.2]

$K = 16$: mean rewards = [0.9, 0.85, 0.8, 0.75, 0.7, 0.65, 0.6, 0.55, 0.5, 0.45, 0.4, 0.35, 0.3, 0.25, 0.2, 0.15]

Let $\Gamma = 2$ or $0.5$. Let $w = 0.9$. The results are shown in Figure 3 below.

2) We also study different $\Gamma$ values in the appropriate distribution that lead to different inference errors. Consider $\Gamma = 0.05, 0.1, 0.2, 0.5, 2, 5, 10, 15$ in the experiments. Let $K = 2$ with mean rewards = [0.7, 0.3]. Let $w = 0.9$.

The results are shown in Figure 4 below.

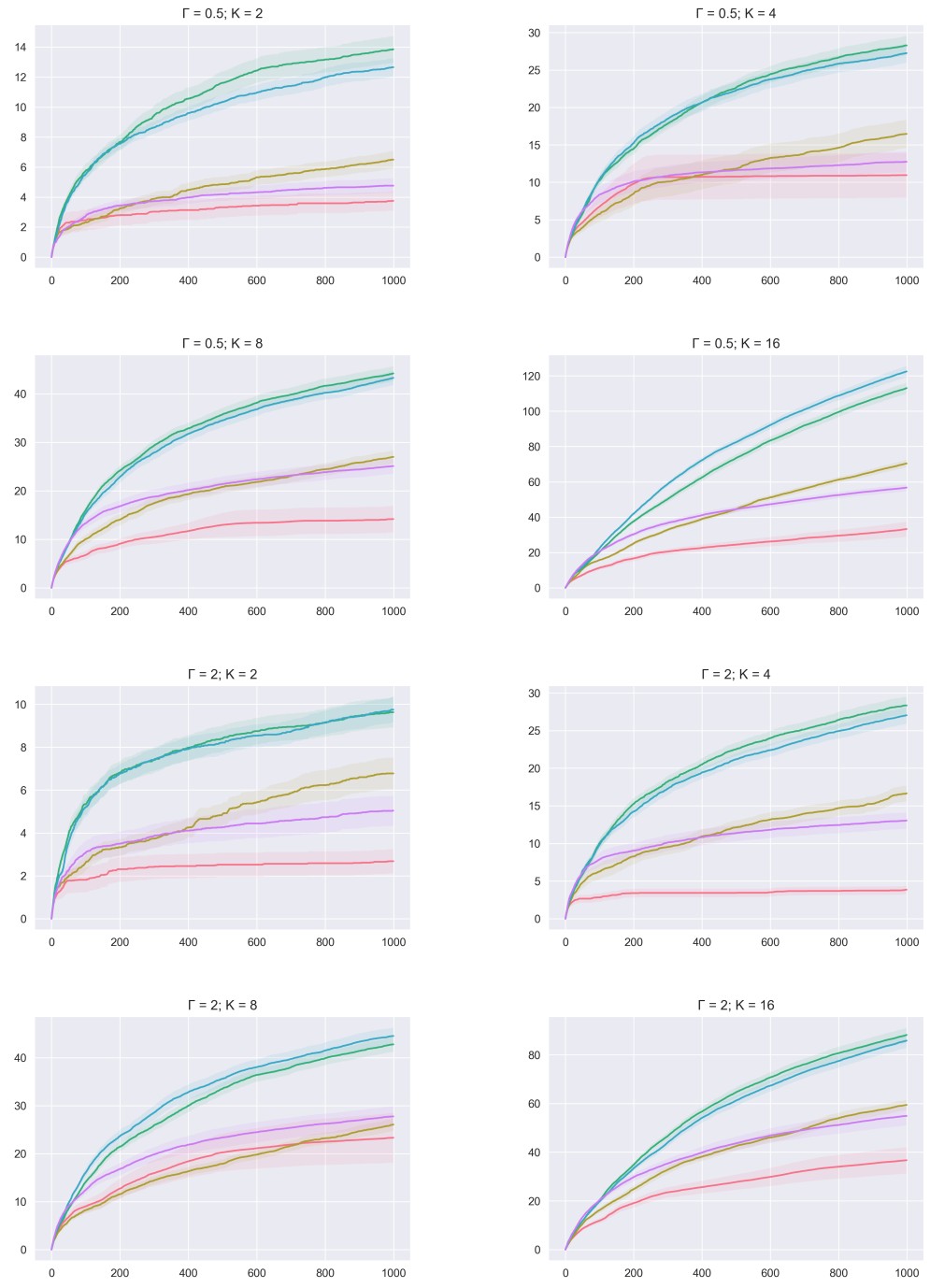

Figure 3: Experiments with different $K$ with $\Gamma = 0.5, 2$. The curve-algorithm correspondence is the same as in Figure 1.

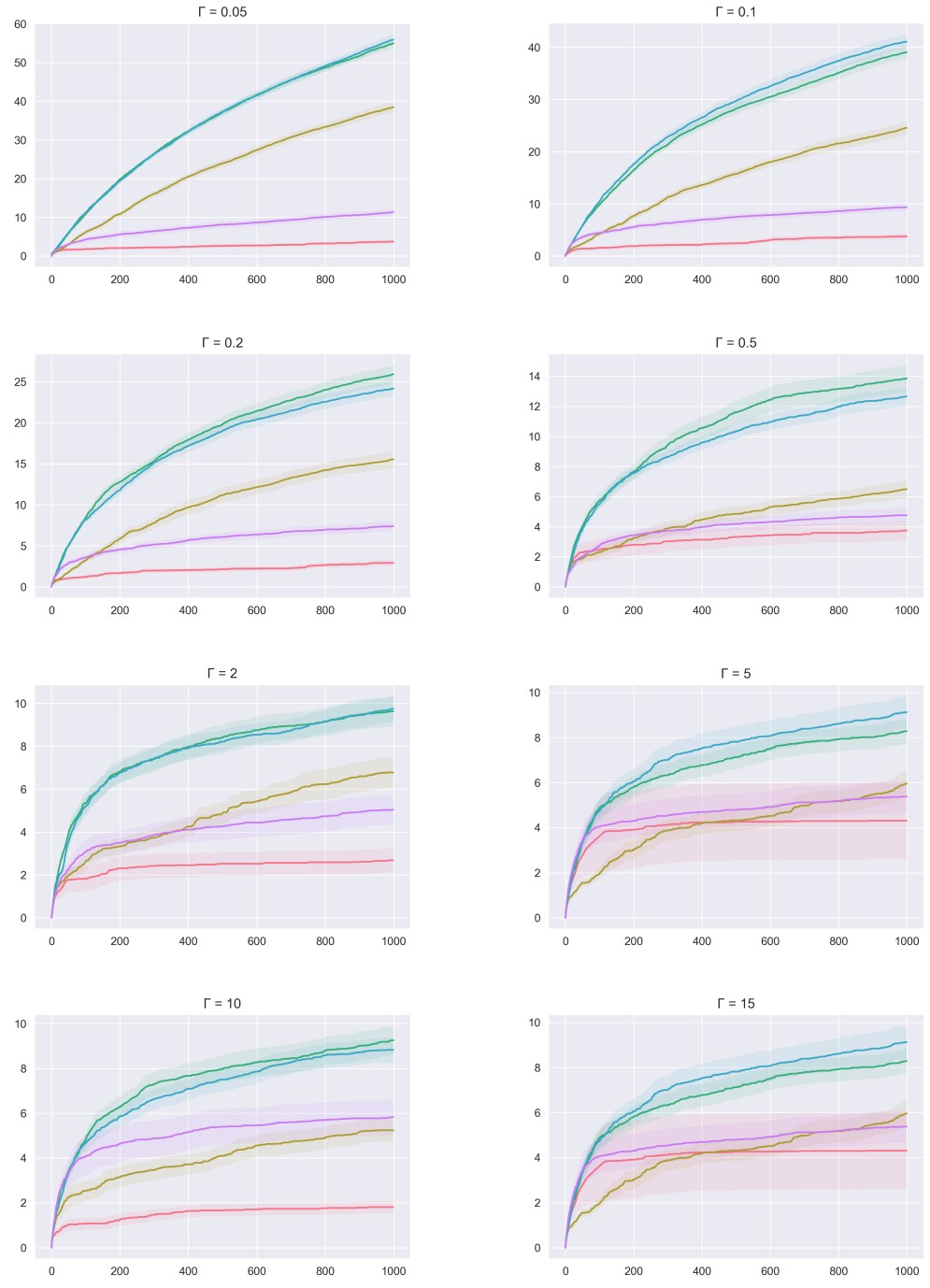

Figure 4: Experiments with different Γ. The curve-algorithm correspondence is the same as in Figure 1.

