# OpenReview forum: "Optimal Regret Is Achievable with Bounded Approximate Inference Error: An Enhanced Bayesian Upper Confidence Bound Framework"
_NeurIPS.cc/2023/Conference — NeurIPS 2023 poster_

### Official Review · Reviewer_tsC9 · 2023-07-04

**Soundness:** 3 good
**Presentation:** 3 good
**Contribution:** 3 good
**Rating:** 7
**Confidence:** 3

**Summary:**

In this paper, the authors consider Bayesian bandit algorithms where exact posterior is not available and
only approximations are available.
The authors prove that if $\alpha_1$-divergence and $\alpha_2$-divergence between the exact posterior and approximation are small,
then a modification of Bayesian UCB (EBUCB) achieves $O(\log T)$ regret.
In the previous work, it is known that Thompson sampling with approximate inference has $\Omega(T)$ regret.
The authors also show negative results for EBUCB and Thompson sampling under the condition on
only one $\alpha$-divergence.
Finally, in synthetic environments the authors confirm their theoretical findings.

**Strengths:**

1. This paper solves an important problem (Bayesian bandit algorithms using an approximation of the posterior) and provides a nice result (the upper bound has $O(\log T)$-regret and the dominant term does not depend on $\varepsilon$, which is surprising).
2. Although the environments are simple and synthetic, they conducted experiments.

**Weaknesses:**

1. In Corollary 3.8, the dependence of $\varepsilon$ is hidden (not discussed).
2. The optimal parameters of the algorithm are not known in practice.

**Questions:**

1. In Corollary 3.8, does the small $o$ notation hides $\varepsilon$? Could you briefly discuss its dependence?

**Limitations:**

Limitations are not discussed.

---

> ### Author Rebuttal · Authors · 2023-08-10
>
> We greatly appreciate the reviewer for the careful reading and valuable comments. We address the reviewer's concerns below.
>
> Q1. "Dependency on $\epsilon$". The regret bound indeed contains $\epsilon$. The exact finite-time upper bound is provided in Step 4 in the proof of Theorem 3.7 (We only show the most dominant term of the regret bound in the main paper to improve the readability). It includes the explicit dependence on $\epsilon$, which is $M_{\epsilon,1}$ and $M_{\epsilon,2}$ there, and obviously the error about $M_{\epsilon,1}$ and $M_{\epsilon,2}$ increase as $\epsilon$ increases.
>
> However, this dependence on $\epsilon$ does not appear in the dominating term. The main reason is that the exact posterior will be more "concentrated" on the true mean with little variability as the time t increases, and the impact from the $\epsilon$ error will vanish. We have some related discussions in Remark 3.9.
>
> To elaborate more, we use a simple example to provide some intuitions. Consider two actions with true mean rewards 0.3 and 0.4. Then after time t, their exact posteriors will be approximately Beta(0.3N_1(t), 0.7N_1(t)) and Beta(0.4N_2(t), 0.6N_2(t)), which is ``concentrated” more and more around the mean rewards 0.3 and 0.4 as t increases. The alpha divergence between Beta(0.3N_1(t), 0.7N_1(t)) and Beta(0.4N_2(t), 0.6N_2(t)) goes to infinity when both N_1(t) and N_2(t) go to infinity (which is true by the information lower bound). Therefore, the $\alpha$-divergence between the exact posteriors of two actions will keep increasing as t increases (finally larger than $\epsilon$). Hence, intuitively speaking, the $\epsilon$ error can only “substantially” impact the regret up to some time step $T_0$, and this additional error is indeed captured by our error bound: You can see the terms $M_{\epsilon,1}$ and $M_{\epsilon,2}$ that depending on $\epsilon$ in Step 4 in the proof of Theorem 3.7. However, they are not the dominating term since after time $T_0$, $\epsilon$ error cannot impact the regret too much as the alpha divergence between Beta(0.3N_1(t), 0.7N_1(t)) and Beta(0.4N_2(t), 0.6N_2(t)) has been sufficiently large.
>
> Q2. "Optimal Parameters". The exact values of $\alpha_1$ and $\alpha_2$ depend on the users’ choice of the Bayesian inference algorithms. Our theoretical results are built upon a general Assumption 3.1 where the Bayesian inference algorithms are not specified.

---

> > ### Comment · Reviewer_tsC9 · 2023-08-18
> >
> > I appreciate the authors for clarifications. I would like to keep the current scores.

---

> > > ### Author Response · Authors · 2023-08-18
> > >
> > > Thank you very much. We appreciate the reviewer for your reading and reply.

---

### Official Review · Reviewer_Fvi2 · 2023-07-06

**Soundness:** 3 good
**Presentation:** 3 good
**Contribution:** 3 good
**Rating:** 6
**Confidence:** 4

**Summary:**

This paper studies Bayesian bandits with approximate inference errors. The authors proposed an algorithm called the Enhanced Bayesian Upper Confidence Bound (EBUCB). Under a two-bounded $\alpha$-divergence assumption, the authors show that EBUCB can achieve the optimal logarithmic regret. The authors also show that with sub-linear regret cannot be achieved with only one-bounded $\alpha$-divergence.

**Strengths:**

The authors provide the first $\log(T)$-type of regret for Bayesian bandits with constant approximation error (under 2-bounded $\alpha$-approximation). This result is obtained based on a novel sensitivity analysis of quantile shift. Both the result and the analysis look interesting to me. The authors also show that, under 1-bounded $\alpha$-approximation, one cannot obtain sub-linear regret; this negative result further justifies the necessity of 2-bounded $\alpha$-approximation (Assumption 3.3).

**Weaknesses:**

My main concern is that the developed results are only for Bernoulli bandits, not for more general distributions (not even for the Gaussian distribution). Can authors comment on why Bernoulli is needed in the current analysis? Or what prevents the derived results from being extended to other distributions?

Another question I have is regarding the constant approximation error for the 2-bounded $\alpha$-approximation (i.e., the $\epsilon$ term in Assumption 3.3). It seems a bit weird to me that the regret bound doesn't depend (too much) on the $\epsilon$ (e.g., in Corollary 3.8). Can authors comment on the reasons behind this?

**Questions:**

See the weaknesses part.

---

> ### Author Rebuttal · Authors · 2023-08-10
>
> We greatly thank the reviewer for the careful reading and valuable comments. We address the reviewer's concerns below.
>
> Q1. "Bernoulli settings and the potential extension to other distributions". One of the major techniques in our analysis is Lemma C.1, which provides tight upper and lower bounds on the tails of approximate distributions to control the quantiles chosen by the EBUCB algorithm. In general, this bound depends on specific distributions, which are Beta distributions plus inference errors in our setting. It is possible to generalize this bound to a certain family of distributions. For instance, Kaufmann, E., [2018] extends the bound for Beta posterior distribution to the exponential family that includes Gaussian (without approximate inference). We believe that by combining Kaufmann, E., [2018] with our techniques in Section 3.2, our results could be generalized to the exponential family with approximate inference. This, however, requires some additional careful technical derivation beyond our current bounds in Bernoulli with approximate inference, which will be our future research direction.
>
> Q2. "Dependency on $\epsilon$". The regret bound indeed contains $\epsilon$. The exact finite-time upper bound is provided in Step 4 in the proof of Theorem 3.7 (We only show the most dominant term of the regret bound in the main paper to improve the readability). It includes the explicit dependence on $\epsilon$, which is $M_{\epsilon,1}$ and $M_{\epsilon,2}$ there, and obviously the error about $M_{\epsilon,1}$ and $M_{\epsilon,2}$ increase as $\epsilon$ increases.
>
> However, this dependence on $\epsilon$ does not appear in the dominating term. The main reason is that the exact posterior will be more "concentrated" on the true mean with little variability as the time t increases, and the impact from the $\epsilon$ error will vanish. We have some related discussions in Remark 3.9.
>
> To elaborate more, we use a simple example to provide some intuitions. Consider two actions with true mean rewards 0.3 and 0.4. Then after time t, their exact posteriors will be approximately Beta(0.3N_1(t), 0.7N_1(t)) and Beta(0.4N_2(t), 0.6N_2(t)), which is ``concentrated” more and more around the mean rewards 0.3 and 0.4 as t increases. The alpha divergence between Beta(0.3N_1(t), 0.7N_1(t)) and Beta(0.4N_2(t), 0.6N_2(t)) goes to infinity when both N_1(t) and N_2(t) go to infinity (which is true by the information lower bound). Therefore, the $\alpha$-divergence between the exact posteriors of two actions will keep increasing as t increases (finally larger than $\epsilon$). Hence, intuitively speaking, the $\epsilon$ error can only “substantially” impact the regret up to some time step $T_0$, and this additional error is indeed captured by our error bound: You can see the terms $M_{\epsilon,1}$ and $M_{\epsilon,2}$ that depending on $\epsilon$ in Step 4 in the proof of Theorem 3.7. However, they are not the dominating term since after time $T_0$, $\epsilon$ error cannot impact the regret too much as the alpha divergence between Beta(0.3N_1(t), 0.7N_1(t)) and Beta(0.4N_2(t), 0.6N_2(t)) has been sufficiently large.

---

> > ### Comment · Reviewer_Fvi2 · 2023-08-18
> > **Response**
> >
> > I thank the authors for their rebuttal. I'd like to keep my current scores and suggest the authors add related discussion into the paper during revision.

---

> > > ### Author Response · Authors · 2023-08-18
> > >
> > > Thank you very much. We appreciate the reviewer for your reading and reply. We will make sure to add these discussions into the final version of our paper.

---

### Official Review · Reviewer_XLXW · 2023-07-07

**Soundness:** 3 good
**Presentation:** 3 good
**Contribution:** 2 fair
**Rating:** 5
**Confidence:** 4

**Summary:**

This paper considers the standard multi armed bandit problem with a prior on rewards, allowing the design of Bayesian algorithms such as Thompson sampling and Bayesian UCB. The problem of interest is when the exact posterior distributions are not available. Rather an approximate posterior is available. It has been known that even with a small constant alpha divergence error between the true and approximate posterior, Thompson sampling does not converge. This paper shows that with a two bounded alpha divergence (see Assumption 3.3) Bayesian UCB achives order optimal regret bound.

**Strengths:**

The problem of designing Bayesian optimization algorithms which find the optimal action in the presence of approximate distributions is very interesting.

**Weaknesses:**

The setting is motivated with the difficulty of obtaining true distributions that often arises in complex models and when using methods such as variationally inference. The results are however proven on a system of Bernoulli distributions where the posterior is available in closed form. I think this break the logic on motivation to a great extent.

While I tried to read all the proofs and details, I could not obtain a good intuition into why small alpha divergence is not enough and two alpha divergence works; how can this be used when applied to for example MCMC or variational inference.

**Questions:**

1. Could authors provide more intuition into the significance of their results for the complex settings where the distributions are approximated. It seems the results are limited to simple settings where the posterior is easily obtained. Could authors intuitively explain why small alpha divergence is not enough and two alpha divergence works.

2. As small alpha divergence is not enough and only two alpha divergence works in this setting, one would expect Theorem 3.7 to fail when $\alpha_1=\alpha_2$. In line 254, it is stated that we may choose $\zeta=\frac{1}{\tilde{\alpha}_2}$. If, in addition, we set $\alpha_1=\alpha_2$, implying $\tilde{\alpha}_1=\tilde{\alpha}_2$, the upper bound in Theorem 3.7 still seems to work. Could authors explain what happens here when $\alpha_1=\alpha_2$ and if the Theorem fails.


**Limitations:**

As mentioned above, the main limitation of the paper seems to be the simple setting, where the posteriors are available in closed form. The results do not seem to be extendable to more complex setting where approximate distributions are actually relevant.

---

> ### Author Rebuttal · Authors · 2023-08-10
>
> We greatly appreciate the reviewer for the careful reading and valuable feedback. We address the reviewer's concerns below.
>
> Q1. "Significance of current results for the complex settings". The key contribution conveyed in this paper is the theoretical insights and guidelines that positively support the practical use of approximate Bayesian inference in bandits. This, to the best of our knowledge, has not been addressed in the previous literature, even in Bernoulli bandit problems.
>
> Our results contribute to understanding the approximate Bayesian bandit methods with complex settings from the following aspects:
>
> 1) Our study takes the very first step in investigating the theory paradox in approximate Bayesian bandit methods. As such, we validate our current framework on a basic problem setting, the Bernoulli bandit problems, and consider the more general bandit problems as our future research directions. Our framework is generic, and could be further extended to more complex settings. Below is a direction to undertake for more complex settings:
> Generalization to the exponential family: Kaufmann, E., [2018] extends the bound for Beta posterior distribution to the exponential family that includes Gaussian (without approximate inference). By combining Kaufmann, E., [2018] with our techniques in Section 3.2, our results could be generalized to the exponential family with approximate inference. This, however, requires some additional careful technical derivation beyond our current bounds in Bernoulli with approximate inference, which will be our future research direction.
>
> 2) Our study provides theoretical support for the superior performance of approximate Bayesian bandit methods, which is not limited to the basic settings. With bounded inference error, Phan et al. [2019] indicated negative theoretical results in multi-armed bandit problems, which contradicts the superior performance of approximate Bayesian bandit methods in practice. To this end, our work resolves this paradox by showing positive results, and further provides direct guidance for real-world algorithm design.
>
> 3) The two $\alpha$ s should be in different regions (one $\alpha$ greater than 1, and the other $\alpha$ less than 0) to guarantee that $P_2$ is close to $P_1$ from both “directions”. As we have discussed after Assumption 3.3, “Intuitively speaking, minimizing $D_\alpha(P_1, P_2)$ when $\alpha$ is large (greater than 1), $P_2$ is flattened to cover $P_1$’s entire support, while when $\alpha$ is small (less than 0), $P_2$ fits the $P_1$’s dominant mode.” Therefore, with one bounded alpha divergence, one can only guarantee that $P_2$ is close to $P_1$ from one “direction”, which could lead to the degenerating performance when using the approximate distribution.
>
> Q2. "$\alpha_1$ = $\alpha_2$". In Assumption 3.3, we explicitly state that two parameters should satisfy that $\alpha_1 > 1$ and $\alpha_2 < 0$. Therefore, Theorem 3.7 excludes the setting where $\alpha_1$ = $\alpha_2$ or $\alpha_1$ is close to $\alpha_2$.
>
> In particular, the two $\alpha$ s should be in different regions (one $\alpha$ greater than 1, and the other $\alpha$ less than 0) to guarantee that $P_2$ is close to $P_1$ from both “directions”. As we have discussed after Assumption 3.3, “Intuitively speaking, minimizing $D_\alpha(P_1, P_2)$ when $\alpha$ is large (greater than 1), $P_2$ is flattened to cover $P_1$’s entire support, while when $\alpha$ is small (less than 0), $P_2$ fits the $P_1$’s dominant mode.” Therefore, with one bounded alpha divergence, one can only guarantee that $P_2$ is close to $P_1$ from one “direction”.
>
> Moreover, in general, if Assumption 3.3 holds for $\alpha_1 <0$ and $\alpha_2 < 0$, even if $\alpha_1$ and $\alpha_2$ are different, we cannot obtain a sublinear regret in Theorem 3.7, where the counterexamples can be similarly constructed as in Theorem 3.12/3.13.
>
> To further address the author’s concern, we will revise and add the following to the introduction section to clarify that two $\alpha$ s should be in different regions:
>
> “However, we will provide a novel theoretical framework and point out that the answer could be 'Yes' when the inference error measured by two different $\alpha$-divergence is bounded where one $\alpha$ is greater than 1, and the other $\alpha$ is less than 0 (which guarantees that the approximate posterior is close to the exact posterior from both “directions”). ”

---

> > ### Comment · Reviewer_XLXW · 2023-08-17
> >
> > Thank you for your response and clarifications. That answers my misunderstanding on the choice of alphas. I still find the contribution limited given that in the Bernoulli case the posteriors are available in closed form and that affects the motivation of the setting.

---

> > > ### Author Response · Authors · 2023-08-18
> > >
> > > We greatly appreciate the reviewer for your reply and for increasing our score. We are glad to hear that our response helped clarify the choice of alphas. Regarding the problem setting, we understand the limitation raised by the reviewer, and we will leverage our techniques in Section 3.2, especially the bounds related to two alpha divergences, to study more general bandit problems and algorithms as our future research direction.

---

### Decision · Program_Chairs · 2023-09-21

**Decision:**

Accept (poster)

**Comment:**

This paper makes a solid contribution, but there was a disagreement on the impact since the paper considers the Bernoulli bandits where the closed form expressions are available. While this is certainly a weakness, I believe the contribution is still meaningful in that it successfully delivers the message that a constant approximation ratio the approximate inference can still lead to a sublinear regret bound. This is worth being known in the community.